



1 **Interactions between climate change and human activities during the Early to Mid Holocene in**
2 **the East Mediterranean basins**
3

*J.F. Berger, Université de Lyon, CNRS, Université Lyon 2-Lumière, UMR 5600 EVS, F-69007, France*

*L. Lespez, UMR CNRS 8591 LGP, Univ-Paris Est Créteil (UPEC).*

*C. Kuzucuoğlu, UMR CNRS 8591 LGP, Université Paris 1-Panthéon Sorbonne.*

*A. Glais, UMR CNRS 6554 LETG-Caen, Univ. Caen-Basse Normandie.*

*F. Hourani, Faculty of Archaeology and Tourism, Univ. of Jordan.*

*A.Barra, Université de Lyon, CNRS, Université Lyon 2-Lumière, UMR 5600 EVS, F-69007, France.*

*J. Guilaine, Collége de France, Paris, France.*

**Key words:** Early Neolithic, PPNB/Monochrome, Rapid climatic change, Northern Greece, Cyprus,
Anatolia, geomorphic impact, mobility, adaptation.
**Abstract**
This paper focuses on Early Holocene Rapid Climate Changes (RCC) records in the Mediterranean
zone, which are under-represented in continental archives (9.2 to 8.2 ka events) and on their impact on
prehistoric societies. This lack of data handicaps indeed assumptions about climate impact on human
societies which flourished in recent years. Key questions remain about the impact of Early Holocene
cooling events on the Mediterranean climate, ecosystems and human societies. In this paper, we
discuss some examples from river and lake systems from the eastern to central Mediterranean area
(Central Anatolia, Cyprus, NE and NW Greece,) that illustrate some paleohydrological and erosion
variations that modified the sustainability of the first Neolithic populations in this region. Results
allow us to present direct land-sea correlations, and to reconstruct regional long-term trends as well as
millennial to centennial-scaled climatic changes. In this context, we question the socio-economic and
geographical adaptation capacities of these societies (mobility, technology, economic practices, social
organisation) during the "Early Holocene" interval (11.7 to 8.2 ka) which corresponds partly to the
Sapropele 1 deposition in the Eastern Mediterranean sea.
**Introduction**
Expected to have had a large impact on past societies, RCC is often considered as one of the main
factors causing socio-economic and cultural changes, migrations, and even collapses (Weiss et
al., 1993, Cullen et al. 2000, Staubwasser and Weiss, 2006, Weninger et al., 2006). According to
this climatic determinism, a RCC would be much harder (if not impossible), for a human society
to adapt to, thus leading to radical societal transformations. In the course of this debate, recent
and ongoing researches on Neolithic societies point to the necessity to focus simultaneously on
(i) the economic, socio-cultural, technological and cognitive transformations of the human group
living on site(s), (ii) the sharpening of old and new chronological series within the site(s), (iii)
the development of contextual analyses associated with geoarchaeological researches, and (iv)
investigating, with a high resolution and multi-proxy approach, the vicinity of Neolithic sites which
are yet poorly studied in connections with the settlements at site. Such approach and methodology are
indeed the most appropriate for reconstructing and interpreting the relationships between
environmental and societal event records which have accompanied (or not) a rapid climate change
and to better estimate adaptability to changing environments. As a matter of fact, a lack of a RCC
signature in the climatic and environmental proxies studied in any sediment record may have
several meanings: an incorrect assessment of a signal, an insufficient chronology control, a
disconnection between the *locus* studied and neighbouring areas where sedimentary archives
would be more favourable for recording a rapid climate change, etc. These are the reasons why
it is often suspected that the absence of signature of a RCC event in continental archives, is more
often due to the low temporal resolution of the available records rather than to the absence of
the climatic signal on the local scale. This problematic situation is now increasingly addressed by
new results focusing on high-resolution analyses and chronologies, as well as on records
associating both the archaeological sites and their surrounding geomorphologic/environmental





archives. In this paper, the goal is to highlight the variety of occurrences of Early Holocene RCC records using (i) interconnected water-related systems (rivers and wetlands) associated with Neolithic sites in contrasted areas of the Eastern Mediterranean basin, and (ii) the characteristics of the main morphogenic and hydrosedimentary responses to RCC on the catchment or lacustro-palustrine scales. We present below four recently-investigated continental fieldwork areas, where new data have been acquired concerning the 9.5 to 7 ka timespan. These data are discussed in the context of their proximity with excavated archaeological sites or with regional cultural trends on the regional scale (Central Anatolia, Cyprus, Eastern Macedonia, Corfu island). Using different spatial scales, from the site to the region and from the eastern to the central Mediterranean, the hydrogeomorphic and ecological impacts of these EH RCCs are evaluated, along with their potential impacts on the first Neolithic societies.

**1. State of the Art**

During the first half of the Holocene, the Eastern Mediterranean regions experienced a climate regime significantly wetter than today, coherently indicated by regional marine and terrestrial isotopes (Bar-Matthews et al. 1997, Roberts et al. 2008, Robinson et al. 2006), the Dead Sea level maximum (Migowski et al. 2006) and the sapropel S1 formation period in the Eastern Mediterranean sea favoured by freshwater high runoff of tropical monsoonal origin (Rossignol-Strick 1999, Rohling et al. 2015). During this period, changes in Mediterranean cyclogenesis would have been potentially influenced by lower Sea Surface Temperature (SST) and evaporation (Brayshaw et al. 2011, Rohling et al. 2015). The general trend toward climate amelioration after the Younger Dryas, favored the development and diffusion of agriculture from nuclear areas in the Near-East (Willcox et al. 2009), the Levant (Bar Yosef, Belfer-Cohen 1989) and Anatolia (Özdoğan, 2011; Kuzucuoğlu, 2014). This Early Holocene phase was nevertheless rhythmed by several pluricentennial abrupt climatic pulsations. Compared to today, the climate was then much more sensitive to freshwater forcing than to solar activity (Teller and Livingston 2002, Fletcher et al. 2013). For example, in the Greenland ice cores, three "rapid events" (RCC) caused by meltwater pulses (MWP) are recorded ca 10.2, 9.2 and 8.2 ka ago, together with at least 11 other similar events documented for the entire Early Holocene (Teller et al., 2002, Fleitmann et al., 2008). In the eastern Mediterranean, extension of the Siberian anticyclone to the Eastern Mediterranean (regular influx of cold air masses) also played a major role during the Holocene period. For example, cold air from the Siberian High (SH) extension created a rapid sea surface (SST) cooling (Rohling et al. 2002) (fig. 1). The multi-centennial variability of the GISP2 terrestrial potassium (K+), a proxy recording the strength and temporality of the SH (Mayewski et al., 1997), shows a stronger SH during some Holocene cold periods in the Eastern Mediterranean, with repetitive impacts on the Anatolian/Aegean areas (Rohling et al., 2002, Weninger et al., 2006, 2014). These latter authors identify a 'RCC-corridor', which runs from the Ukraine, through south-eastern Europe, into the Aegean and large parts of Anatolia and the Levant, as well as onto the islands of Cyprus and Crete. Rogers (1997) linked cyclogenesis in the Mediterranean with positive (strong) SH anomalies, while eastern Mediterranean flood activity shows periodically a positive relationship with an increasing trend in the K+ proxy (Benito et al. 2015).

The potential impact of the 9.2 ka abrupt climatic event on human societies during the Neolithic "revolution" has rarely been explored, in any case much less so than the 8.2 ka event. In this debate, the effects of the worldwide "8.2" climatic event on the Mesolithic and Neolithic societies have been under discussion for a decade, with interpretations varying from abandonment of sites to collapse, from large-scale migration to sustainability of occupation and social adaptation… (for a complete overview see Gehlen and Schön, 2005, Staubwasser and Weiss, 2006, Weninger et al., 2006, 2014, Berger and Guilaine, 2009, Flohr et al. 2015). Climatic records show that the 8.2 ka event resulted in some of the most extreme environmental perturbations of the Holocene. For this reason, it has been subject to an abundant literature since being first discussed (Alley et al. 1997). Extended over a time-span of 100-150 years in GISP2-GRIP polar archives (Thomas et al. 2007), its duration has been found longer in numerous marine and continental proxies (fig01). In the eastern Mediterranean and other regions, the RCC interval between 8.6 and 8.0 ka spans a longer time period than in the ice record,





supporting the idea of an enhanced Siberian high-pressure anticyclone over Asia (Rohling and Pälike,
2005, Weninger et al. 2014) controlling a global intensification of atmospheric circulation with cooler
temperatures in polar regions (Mayewski et al., 2004) and drier and cooler conditions in the
Mediterranean basin (Rohling et al., 2002; Bar-Matthews et al., 2003; Fletcher and Zielhofer, 2013;
Gómez-Paccard et al., in press). Meanwhile, pollen and SST data have been increasingly studied in
marine mediterranean archives for 15 years.
In parallel to these ice and marine records, Mediterranean basin scaled continental records reveal a
paucity of evidence of Early Holocene RCC. For examples, Berger (2015) and Berger et al. (in press)
underline episodes of lateral mobility/erosion of rivers and successive entrenchments of active beds
although the period is dominated by a multi-millennial-long predominance of pedogenic processes.
Although Early Holocene earth-surface processes are rarely documented in clear geomorphological
and chronological frameworks from the Southern Levant, there is some evidence for abrupt
geomorphological responses in the most fragile (semi-arid) regions during Holocene RCCs (Cohen-
Seffera et al., 2005). But there is a general lack of very precise geomorphological studies for this
period (Berger and Guilaine 2009, Zielhoffer et al. 2008, 2012).
Divergent information from different proxy records and chronological uncertainties are often major
limitations to our understanding of abrupt climatic changes and their impact on continental
environment (Desprat et al. 2013). Early Holocene palaeoenvironmental data derive first from inferred
changes in lake hydrology (isotopes and salinity changes, water level variations; Magny 2004,
Eastwood et al. 2007, Roberts et al., 2008 and 2011; Kuzucuoğlu et al. 2011), quantitative pollen
studies (Eastwood et al., 1999; Roberts et al., 2001; Pross et al. 2009, Peyron et al. 2011, Bordon et al.
2009), fire analysis (Vanniere et al. 2011), and also from cave speleothems records (Bar-Matthews et
al. 1997, Verheyden et al. 2008, Göktürk et al. 2011, Frisia et al. 2006) and marine cores (Kothoff et
al. 2008, Combourieu-Nebout et al. 2013, Desprat et al. 2013, Fletcher et al. 2012, etc.) (fig02). Multi-
proxy comparisons (pollen-inferred changes in plant functional types *vs* modern analogues), help
identifying a strong connectivity with the Mediterranean watersheds, in particular when deciduous
woodland switches to sclerophyllous woodland and scrub, or when mountainous assemblages increase
during colder events (Peyron et al. 2011, Combourieu-Nebout et al. 2013....).
Despite these many recent paleoclimate studies, it is still difficult to imagine the relationships between
climate and hydrogeomorphology in the eastern part of the Mediterranean basin at the secular scale
during the Early Holocene. For example, is it possible to consider a synchronous and similar
hydroclimatic and geomorphic functioning all through the area from the Ionian-Aegean basin to the
Levant regions? Is there a latitudinal climatic barrier between a north part and a south part of the
eastern Mediterranean, as there is between the central and western Mediterranean (Magny et al.,
2013)? How much seasonal or annual water is available for soil and vegetation, notably during the
main RCCs ? What links can be found between changes in practices or in population movements, that
may be connected to past hydrological changes in the continental areas?
An archaeological laboratory dedicated to vulnerability research in prehistoric periods is ongoing
(Clare and Weninger 2008, Bocquet-Appel et al. 2014, Borrell et al. 2015, Flohr et al. 2015, "2010-
2020" Paléomex project)..., looking for the widest possible field of alternative societal modes and
responses to environmental changes/versus natural hazards. The RCCs-mechanism and their
millennial cycles during the Holocene give opportunities to study the impact of rapid events on
cultural transitions and/or migrations/mobility, and to explore the societal adaptibility modes in stress
conditions through time and in specific contexts. The current main hypotheses are based on regional
chronocultural patterns defined by Cumulative Probability Density Function (CPDF) techniques on the
one hand, and on the time parallelism between a decrease in Radiocarbon date clusters and the
assertion of a RCC on the other hand. As proposed by Flohr et al. (2015), a more critical approach is
now clearly needed to better characterise socioenvironmental relations with climate and environmental
changes during RCC, an approach that would be more trustful than the use of regional [14]C-dates series
which may be neither rigorously quality-checked nor solidly correlated in space and time. In order to
face the need for highly constrained dating strategies in archaeological contexts, the intra-site scale is
now being applied in sites such as Çatalhöyük (Clare and Weninger 2008, Marciniak et al. 2015),
Aşıklı Höyük (Stiner et al., 2014), Tell Sabi Abyad in the Upper Euphrates (van der Plicht et al. 2011,
Akkermans et al. 2014)...etc. Extensively applied in stratigraphy as well as in space at the site, this
approach aims to establish continuity/discontinuity in occupation and cultural changes within a



sensitive timing. For example, Clare and Weninger (2008) and van der Plicht et al. (2011)
demonstrated that a multiplication of $^{14}$C dates by CPDF at a single site, can fill or confirm the
suspicion of a hiatus. A critical analysis of the state of regional radiocarbon databases is therefore
essential, not only, as recently applied by Flohr et al. (2015), with a selection of shortlived dates, but
by a systematised intra‑site stratigraphic and taphonomic evaluation such as that recently conducted on
the Dikili Tash and Sidari sites (Greece) (Lespez et al., 2013, submitted; Berger et al. 2014). We
consider that this approach is the most reliable way to observe the degree of continuity of human
occupation and thus to establish its possible links to local hydrogeomorphological dynamics during
RCCs. But such archives are rare, and primarily dependant on the site position in the catchment area,
on the proximity of the site with favourable sedimentary archive areas (like floodplains, swamps, foot
slopes...) and on the type of site (tells being less favourable to hydrosedimentary records as soon as
they emerge from the floodplains). In addition to a lack of $^{14}$C dates on site, the lack of archaeological
data corresponding to the same timing as a rapid and short‑lived event, may have other causes than the
absence of a link: a prevailing theoretical bias, old wood effects (while dates on charcoal have long
been privileged, seeds and other short-lived organic matter are preferred), restricted excavation of site
surfaces and periods...
Not only many palaeoclimate and environmental records have neither sufficient temporal resolution
nor chronological precision, but the sensitivity of a continental record to detect a decadal‑scaled
climatic anomaly is also rarely assessed. For this latter factor, more detailed geographical and
bioclimatic local frameworks within regional assessments are needed. The availability of such
assessments is necessary for discussing not only the local impacts of climate events on the resources
and landscapes (Clare and Weninger 2010), the societal impact or non-impact of a RCC (Roberts et
al., 2011; Kuzucuoğlu, 2015), and our knowledge of past adaptation strategies (Berger 2006; Berger
and Guilaine 2009; Lespez et al., 2014, in press; Flohr et al. 2015). As far as the study of early farming
societies is concerned, data about micro‑regional and local effects of RCCs will usefully replace or
complete, as far as the study of early farming societies is concerned, the information delivered by the
key regional – and remote – climate references which are regularly called for in research papers
(glacial, marine, continental dendrochronological series, speleothems...) (Weninger et al. 2006, 2009,
Kuzucuoğlu, 2009). Local detection of RCC impacts are still too rarely attested to on archaeological
sites or in continental river archives close to sites occupied by the first farmers or the last hunter-
gatherers (Berger and Guilaine 2009, Zielhoffer et al. 2012, Lespez et al. 2013, Berger et al. in press).
We thus propose here a "bottom-up approach" of the impact of climate changes on the Early Neolithic
societies. We intend to demonstrate that precise geoarchaeological investigations in Neolithic sites,
when based on systematic stratigraphy studies, rigorous radiocarbon series and on a contextual
archaeological approach, end up proposing new socioenvironmental schemes on the local scale.
Meanwhile, we explore new hypotheses about the impacts of the Early Holocene RCCs on the
environments as well as the responses of Neolithic societies.
**2. Material and methods: new continental data with high chronological resolution in the centre**
**and east of the Mediterranean basin**
2.1. Central Anatolian and Cyprus cultural contexts
These two regions neighbour the nuclear areas of the Pre-Pottery Neolithic A (PPNA) (11.7-10.5 ka)
in the Levant and of SE Turkey (middle and upper Tigris and Euphrates valleys). Where identified (in
the Levant, SE Turkey, Iran, Cyprus, central Anatolia), the "Pre-Pottery Neolithic" (PPN) corresponds
to a "Neolithisation" period during which packages composed of several or all characteristics of the
Neolithic are identified in excavated settlements: sedentism, housing, pre‑domestication (followed
possibly by domestication) of sets of plants and/or animals (Fuller et al., 2011, Zeder, 2011; Stiner et
al., 2014), symbolism, art, social organisation and ritual behavior (Cauvin 2002 ; Simmons 2011).
Increased sedentism and plant and animal domestication practices are asserted during the period of
relative climate stability that follows rapidly the turmoil of the Holocene onset warming up and its
consequences on the vegetation and water resources. This has greatly contributed to conceiving the
Neolithisation processes in the Near East as an incremental continuum (including several and distinct
successful and unsuccessful attempts: Willcox et al., 2012) in disconnected "cores" spread over the



region, with relatively minor disruptions (Borrell et al. 2015). Recently, a major cultural discontinuity
has been observed in the archaeological PPN records of the northern Levant, that lasted from 10.2 to
9.8 ka and was followed by a substantial cultural transformation indicating a break in the
Neolithisation process (Weninger et al. 2009, Borrell et al. 2015). This early discontinuity corresponds
to a hiatus in settlements, which covers almost the totality of the time span traditionally attributed to
the Middle PPNB in the Levant (10.2 – 9.6 ka) (Borrell et al. 2015). In Cyprus, a cultural change is
initiated ca. 9.6/9.5 ka (emergence of the Khirokitia culture: Le Brun et al. 2009). In the
Shillourokambos site (fig02), the change occurs in the early C phase, initiating a different cultural
package which lasted the 2nd half of the 10th mill. cal BP. The cultural change is visible in the quick
decline of the beautiful lamellar tools obtained in the previous phase by bipolar knapping (a strong
PPNB marker in the Levant), replaced by productions directed towards robust pieces (thick and
irregular blades, pikes, sickles with parallel hafting to the edges) (Briois, 2011). Meantime, there is a
decrease in grinding instruments (Perrin 2003). Imports of Cappadocian obsidian collapse. The habitat
reduces in size, concentrating in the southern part of the site. Building materials evolve with the
abandonment of the proto-brick for mud-building techniques. From 9.2 ka on, sheep husbandry plays
an important part, perhaps in association with the development of pastoralism (Vigne *et al*. 2011).
These cultural and economic changes have never been confronted with climato-environmental
evolutions, in spite of their synchronicity with a first global signal (fig01).
In central Anatolia, after the abandonment ca 9.5 ka of early PPNB sites in the Konya plain
(Boncuklu, Can Hasan III) and Cappadocia (Aşıklı), younger PPNB sites appear at other locations : ca
9.6/9.5 ka in Cappadocia (Musular site), and 9.4/9.3 ka in the Konya plain (Çatalhöyük East). This
butchering-specialized site is abandoned ca 9.0 ka, before the apparition of the pottery. From the west
of the Konya plain to the Lake district where sites are founded ca 9.2 ka without pottery (PPN) as in
Bademağacı, and to the Aegean Anatolia (Ulucak), Neolithic occupation continues with no hiatus onto
and during the Early Neolithic period which starts quickly, ca 9.0/8.9 ka, with appearance of pottery.
Pottery appears also with a very similar timing in many other sites in Cappadocia (eg. Tepecik-Çiftlik;
Aşıklı too, possibly…) to the Mediterranean (eg. Yumuktepe) and the Aegean (eg. Yeşilova etc.)
(Fig02, and references herein, especially in Özdoğan et al., 2012a and 2012b). New results (eg. articles
in Özdoğan et al., 2012a, 2012b; Stiner et al., 2015) and from on-going syntheses (eg. Özdoğan, 2011;
Kuzucuoğlu, 2014) suggest that a long-distance neolithisation dynamics originated out of a core
located in Konya plain and Cappadocia. This diffusion arrived in the Aegean region ca. 9.1-9.0 ka
(Özdoğan, 2011). In the Near-East as well as in central Anatolia, Flohr et al. (2015) show that $^{14}$C
dates-based spatio-temporal reconstructions of sites distributions, do not provide evidence for
widespread migrations ca. 9.2/9.0 ka. As a matter of fact, in Anatolia the apparent westward-
progressing cultural influences do not mean automatically "departure" or "migration" from the large
plains ca 9.2/9.0 ka, but rather "diffusion" (Kuzucuoğlu, 2014). For example, the typical "highly-
populated and densely-built large PPN "villages" of Cappadocia (Aşıklı) and Konya plain
(Çatalhöyük-East) do not exist anywhere else nor afterwards. In addition, the earliest Pottery Neolithic
layers (continuing PPN) in the Lake District are culturally distinct from the contemporaneous ones in
the Konya Plain located east (Duru, in Özdoğan et al., 2012b). Archaeological records that, even with
Late PPN/Early PN starting early in the Konya Plain and Cappadocia, there is no direct influence from
there during the transition to the Neolithic and during the Early Neolithic in the Lake Districts (Fig02).
In addition, in western Anatolian, Early Neolithic cultural material from sites occupied at the
beginning of the 9th mill. records the mixing of local traditions with other cultures from the Near East
(diffused along the sea shores?) as well as from the Lake District (diffused westward?) with, again, no
influence from the "core area" in central Anatolia (Konya Plain, Cappadocia).
Consequently, any approach which aims to understand the relationships between climate and human
societies during the time of the Neolithic development and expansion in Anatolia (Kuzucuoğlu, 2014)
must take into account the regional dimension of the economic, technological and social
characteristics of the Anatolian Neolithic, especially in the plains and plateaus of central Anatolia
(Özbaşaran, 2011).
2.2. Northern Greece: cultural and archaeological contexts
The tell of Dikili Tash is located in the south-eastern part of the Drama plain, in eastern Macedonia,



northern Greece (fig02). It is one of the largest tells in northern Greece, covering an area of ca 4.5 ha, with its highest point standing at ca 15m above current ground surface. A freshwater spring lies immediately to the north-east of the tell, and it opens on a large swamp to the south (Tenaghi-Philippon) about which many environmental studies have been published. Ongoing excavations have provided a good insight into the long stratigraphic sequence of this settlement from the bottom of the plain, completed by coring surveys in the deeper humid zones at the southern periphery of the site (Lespez et al. 2013; submitted; Glais et al., 2016). The deepest archaeological l level, very close to the natural soil (a brown leached soil), has been dated 8.54–8.38 ka, ie Early Neolithic.

In Sidari (NW Corfu island), the archaeological excavation revealed in a deep small valley filling an initial Neolithic with red monochrome ceramics, domestic fauna, cereals and mud houses, whose economic status remains to be specified from the ongoing monographic publication of the French-Greek team (fig02). Together with Odmut (Bosnia and Herzegovina) and Konispol cave (Albania) (Sordinas 2003, Forenbaher et Miracle 2005), Sidari was originally considered as one of three sole sites in NW Greece and southern Adriatic area with an apparent Mesolithic/Early Neolithic stratigraphic continuity. On the basis of a new contextual geoarchaeological study (Berger et al. 2014), we recently discussed this aspect, refuting the original interpretation made by Sordinas (1966, 1973).

## 3. Results

The results of the local investigation in the four selected studied are presented from east to west following the Neolithic expansion.

### 3.1. Central Anatolia

Questioning the role of climate on the Neolithic dynamics in central Anatolia from PPN to PN and during the Early PN during the 1st half of the 9th millennium cal BP, means that we have to define the climatic context and evolution from 9.5/9.4 ka to 9.2/9.0 ka. A similar question concerns the transition phase between PN and Chalcolithic ca 8.2-8.0 ka in Anatolia, although many archaeologists suspect the latter distinction between "Neolithic" and "Chalcolithic" to make no sense in Anatolia. Instead, the cultural turning-break that occurs through Neolithic Anatolia ca 8.6 ka, is much more distinct than changes happening ca 8.2/7.8 ka (Düring, 2011; charts in Özdoğan 2012a, 2012b). Nevertheless, the parallelism between cultural changes and the timing of the "9.3" and "8.2" ka RCCs suggest that there may have been a relationship between climate and cultural changes during the events.

The wide and endorheic plains of central Anatolia (Fig02 and 3) open in steppic plateaus ca. 1200 to 1300 m altitude. The altitudes of the three main plains are ca 920 m a.s.l. (Tuz Gölü, to the north), 1000 m a.s.l. (Konya and Ereğli, to the south), and 1050 m a.s.l. (Bor, to the east). In these plains, the current climate is semi-arid with mean annual precipitation ranging from 280 to 340 mm/yr (respectively Konya and Tuz Gölü plain, southern Cappadocia lowlands). This semi-aridity contrasts with the fact that, ca 11.3 ka on, the most ancient Neolithic sites of Anatolia are founded in these plains (Baird, 2012 ; Özbaşaran, 2011), in a timeframe similar to that of the PPN (Pre-Pottery Neolithic) in the Tigris headwaters (Özdoğan, 2011). Results from geomorphologic, geoarchaeologic and palaeoenvironmental researches during the 1990s in the Konya plain (Kuzucuoğlu et al., 1997, 1998, 1999; Fontugne et al., 1999 ; Roberts et al., 1999), in the Tuz Gölü plain (Naruse et al., 1997 ; Kashima, 2002), and more recently in the Bor plain (Gürel & Lermi, 2010 ; Kuzucuoğlu 2015 ; Matessi et al., in press) today allows us to propose a chronological synthesis of the environmental context of the cultural dynamics between the 10th and the 7th millennium cal BP.

The palaeoenvironmental records in the three closed plains of central Anatolia (Fig04) show evidence of alternations of humid and dry phases during the Holocene. The chronological comparison between these phases and the global climatic record shows that, (a) there is a high variability of records in the humid areas sensitive to even slight changes in humidity; (b) some RCC have no correspondence in the environmental records; (c) when a signal occurs in parallel with one of the RCC, the signal varies in nature and magnitude (soil signaled by roots and vegetation, emersion out of wetlands, drying-off, drought, etc). The comparison between the locations of the sediment archives in such an evaporation-sensitive context as that of the central Anatolian endorheic plains shows that the geomorphologic settings of the records (cores and sections) control the signal, ie the type and sensitivity of the drying/wetting wetlands: sub-surficial water in alluvial fans, marshes fed by springs at the external





edges of alluvial fans, springs along faults, karstic outflows, ice and snow-melt from highlands, rivers
etc… (Fig04). Both the topographic specifities of the ecosystems, and the spatial variability of the air
masses transporting humidity in the area contribute to the importance of the regional and local scales
in the palaeoenvironmental records.
According to these records, the general environmental evolution in the region during the Early
Holocene is the following (Fig04):
- After the onset of the Holocene ca 11.4 until 9.5-9.0 ka, springs and rivers in the Konya plain collect
water originating in precipitation and snow/ice melt in the Taurus. This water is also discharged by the
karstic network of the range. This water accumulates into shallow depressions stretching at the foot of
the Taurus along the Konya-Ereğli-Bor plains. For example, the expansion of the Akgöl backswamps
at the southern border of the Ereğli plain (Bottema & Woldring, 1984) is such a signal of a humidity
rise triggered from the Taurus highlands.
- Towards 9.5 ka, alluvial fans start to expand over the LGM marls forming the Konya plain bottom
(Çarsamba and Karaman rivers: Boyer et al., 2006), as well as in the Çiftlik plain up in the
Cappadocian volcanoes (Kuzucuoğlu et al., 2013). This river dynamics-related change is the only
possible signal of a climatic change contemporaneous with the 9.3 ka RCC. This signal is produced by
a change in run-off indicating a rise in spring water and a possible increase in seasonal temperature
contrast. Such a change would have produced enough snow and ice meltwater to initiate the growth of
Holocene alluvial fans over the plain bottoms. During this period, the Adabağ pollen record is marked
by the expansion of an arboreal vegetation dominated by deciduous *Quercus* (Bottema & Woldring,
1984). This alluvial fan initiation corresponds to the abandonment of PPN sites in Cappadocia (Aşıklı)
and Konya (Boncuklu, Can Hasan III). One or several centuries later, Late PPN sites (Çatalhöyük-East
in Konya; Tepecik-Çiftlik in Cappadocia) are founded at locations close to the expanding alluvial fans.
- The soil dated 9.0-8.9 ka in the Adabağ core possibly marks the end of the period of change which
started ca 9.5 ka. With the exception of the Çarsamba fan which continues to grow until 8.6 ka, the
absence of sediment record dated first half of the 9[th] millennium cal BP suggest that the plains were
dry, with little or no water input from the central Anatolian highlands (Cappadocian volcanoes).
- The second half of the 9[th] millennium cal BP is characterised in Konya plain by the interruption of
the torrential dynamics in the Çarsamba fan between 8.6-8.2 ka. During this period, the marshes along
the edges of the Altunhisar fan in the Bor plain seem to have dried off too, although not for as long
since they are well watered (lakes and backswamps) before 8.2 ka when they dry up again. In a
generally dry 9[th] millennium cal BP in central Anatolia, this dry/wet/dry alternation in the northern
shores of the Bor plain (Bayat and Kayı cores), as also the continuing record at Adabağ (fed by Taurus
karstic waters), correspond to local signals.
- The 8.2 ka RCC is present in central Anatolian records as a one century-long dry signal interrupting
backswamps and lakes around the Altunhisar fan between 8.1 and 7.9 ka.
- The most humid climatic phase in central Anatolia starts ca 7.9 ka, and will last until ca 6.5 ka which
marks the beginning of the mid-Holocene dry phase (Kuzucuoğlu, 2015; Matessi et al., in press).
3.2. Khirokitia (Cyprus)
Khirokitia is a Cypriot Late Pre-pottery Neolithic village dated to 8.6-7.5 ka (Le Brun et al. 1987, Le
Brun & Daune-Le Brun 2009). The site is located on the southern foothills of the Troodos Mountains,
at about 6 km from the Mediterranean shoreline (fig05a). It occupies the flanks of a limestone rocky
mound (around 216m above sea level), bounded to the north and east by the Maroni River channel
(Fig05b). At the present time the river channel is ephemeral and forms a rather deep and narrow valley
cut down through a terrace series of Quaternary conglomerates and older fluvio-marine deposits. The
stratigraphic sequence of the site comprises two major series of occupational levels; the articulation
between which, dated to nearly the end of the seventh millennium cal. BC (around 8.2 ka), is marked
by a redistribution of the village space in form of shift and contraction and by a noticeable change in
the botanical and zoological records (Le Brun & Daune-Le Brun 2010; Le Brun *et al*. in press).
Detailed geoarchaeological investigations were performed, mainly at the foot of the eastern slope of
the site, where the archaeological remains meet the river, and on the surrounding river deposits
(Hourani 2008). Results from this research allowed recognition of at least two major sedimentary
events that occurred during the occupation of the site.



The first of these events is a major channel incision concurrent with torrential stream discharges (fig06). It is marked at the foot of the eastern slope by the deposition of a 3.5m thick layer of densely packed, non-sorted, rolled stones and gravel at the base and more stratified but relatively fine-grained gravel and sand near the top. Deposits here underlie the archaeological remains in this sector and unconformably overlie Miocene fluvio-marine sediments. One feature of note is the presence of Neolithic stone tools as well as charcoal lenses, ash and fine fragments of burnt bones and mud brick within the alluvial discharge near the top. A radiocarbon date obtained on ash specks from this unit indicates an age of 8.518 ± 55 year BP.

The second and more prominent sedimentary event is a substantial erosional episode. It is particularly visible in the middle of the archaeological sequence overlying deposits of the first sedimentary event at the foot of the eastern slope (fig06). A 0.6 to 0.8m thick stratum of angular limestone gravel and other archaeological debris divide the 4 meter-high archaeological sequence in this area into two parts. Archaeological structures of the lower part are deeply gullied and appear to be less preserved than in the upper one. Two radiocarbon dates, obtained on charcoal lenses from the debris of two superimposed houses sited on top of the erosional layer, propose respectively the ages of 8.276 ± 55 and 8.248 ± 53 year BP. To this later episode of erosion and surface flows might also be attributed a 3m thick sequence of intersected clusters of alluvial discharges and of side gully debris observed on the river section, slightly upstream of the studied archaeological sequence and opposed to it (fig05b). Here alluvial deposits are composed of loosely packed and unsorted stones, gravel and coarse sand. Gully debris, triggered from the surrounding slopes, are more represented near the base of the sequence where they consist of compacted whitish to dark grey loam, mixed with small white angular and black rounded stones along with flakes of flint, bone fragments and lenses of charcoal. This gully debris was radiocarbon dated to 8.105 ± 55 year BP. The top of the sequence is capped by alluvial dark grey sandy silt, 0.8-1.2m thick, and then by grey-brown loam indicating subsequent decrease in the energy of flows. However, incision during the Late Holocene led to the lowering of the river channel bed, producing a suite of at least two younger river terraces in the area.

The two sedimentary events described above indicate that the region of Khirokitia experienced strong modifications in the hydro-geomorphological configuration around 8.5 and more particularly 8.1 ka. The morphological distinction between these two events, and what could have been the situation before, is difficult to establish adequately in such a dissected area as older terraces are obscured by younger sedimentation and erosion. However, the nature and the extent of the events observed indicate erratic and heavy rainfall conditions that in all probability seem to have occured on a wider regional scale. Not far from Khirokitia, down cutting by 6m was followed by a period of aggradation and alluviation between 8.3 to 7.9 ka in the Vasilikos Valley near Kalavasos (Gomez, 1987)(fig05a). A similar sequence was also observed in the Middle Jordan Valley (Jordan), where marshy deposits corresponding to the beginning of the Holocene were deeply truncated and then recovered after by the red soils associated with the first settlers of the Late Neolithic period (Hourani & Courty, 1997; Hourani 2005; 2010) (fig02).

Notwithstanding man's role in the weakening of the soil cover, neither tectonic activities that may also have facilitated the incision of the riverbed and (or) changes in the direction of the stream runoff as well as lowering of the riverbed both indicate that the Neolithic landscape at Khirokitia resulted predominantly from climatic factors. At Khirokitia, if this period of surface erosion and torrential discharges were to be integrated into a wider regional or global scale, it might then be seen as a regional expression of the worldwide-identified 8.2 ka event. Here, the first cultural implication that can be drawn from this erosional event is the shift and contraction in the village space along with the major changes observed in the botanical and zoological records towards the end of the seventh millennium cal. BC. The attribution of the end of the PPN occupation at Khirokitia to the 8.2 event (Weninger *et al*. 2006) thus cannot be sustained.

3.3. Eastern Macedonia

In Eastern Macedonia, investigations have been developed on the edges of Tenaghi-Philippon marsh. This large marsh located in Northern Greece has been subjected to numerous paleoenvironmental research (Wijmstra et al., 1969; Greig and Turner, 1974; Tzedakis et al., 2006; Pross et al., 2009, Peyron et al., 2011) which constitute reference records for the environmental history of the Eastern





Mediterranean area (fig07). The results of these studies have been focused mainly on climate impact on vegetation cover. In order to track the climatic changes but also the impact of the Neolithisation process, which is here dated from 8.5 ka onwards (Lespez et al., 2013), palaeoenvironmental investigations have been developed from the archeological site to the marsh. Geomorphological research has been focused on the tell and its surroundings (Lespez et al., 2013, submitted) while Pollen and Non-Pollen Palynomorphs (NPP) analyses come from core Dik12, at the bottom of the site, and Dik4 located 2km to the southwest on the edge of the Tenaghi-Philippon marsh (Glais et al., 2016). This core is 3m long and the sediments are mainly constituted by grey to black organic clay. The chronology based on 11 AMS Radiocarbon datings.

The pollen records (fig08) indicate a general decrease in steppe taxa (Artemisia and Chenopodiaceae) and the steady increase of other herbaceous plants such as Cichorioideae, and other ruderal taxa suggesting a return to more humid conditions at the end of the Younger Dryas (ca. 11.7 – 10.2 ka). This is also supported by the recorded appearance of lime trees, an increase of NPPs indicative of eu-mesotrophic conditions and a slight but continuous deciduous oak expansion. These observations are consistent with the regional climatic model (Kotthoff et al., 2008; Peyron et al, 2011). Around 10.2 ka the pollen indicate a gradual and long-term change with great development of arboreal vegetation and the decline of open vegetation cover (AP/NAP ratio increases from 20% to more than 50%). Wetter and warmer conditions have favoured the expansion of all broad-leaved trees, such as oaks, alders and subsequently the appearance of mesophilous taxa such as ostryas, birches, ulmus and evergreen oaks. After a delay in comparison with western Greece (Lawson et al., 2004), it indicates the onset of interglacial conditions. In this context, the first macrocharcoal peak extended (10.6-9.3 ka) corresponds to the biomass development in a still incomplete wooded landscape. Forest expansion was punctuated by a short-term centennial-scale dryer climatic events (9.6-9.3 ka) distinguishable at regional (Kotthoff *et al.,* 2008) and local scale by the increase of xerothermophilous taxa and evergreen *Quercus* (Glais et al., 2016).

After 9.3-8.7 ka, the vegetation cover is marked by a peak of deciduous oaks, the appearance of fir on the top of surrounding mountains, the decrease of Poaceae, Aster type and Cichorioideae taxa and the retreat or even disappearance of woody species limited to Mediterranean contexts. This spread of forest cover was interrupted around 8.7-8.3 ka. The decrease of trees and increase of herbs could indicate the impact of the 8.2 ka RCC but this period also shows the first signs of human impact in the Early Neolithic. They are certainly due to the Early Neolithic settlement implantation in Dikili Tash (Lespez et al., 2013; Glais et al., 2016) benefitting from pristine forested environment with multiple available resources. This is attested to in the NPP record, by a first coprophilous species peak, but also by a decrease of deciduous forest species and increase of herbaceous taxa on the edge of the marsh. Furthermore, at the bottom of the site (Dik 12), high- percentage cereal pollen (around 9% at 8.4 ka) and the increase in ruderal taxa make it clear the anthropogenic impact on vegetation cover associated with agropastoral activities.

Nevertheless, the conjunction with the 8.2 ka event well established at the regional scale a few decades after makes the interpretation more complex and other causes can be evoked to explain the pollen and NPP records. The high percentage of hydro-hygrophytic taxa on the edge of the marsh suggest a contemporaneous rise in the water table level in a drier period well assessed at the regional scale (Pross et al., 2009). Furthermore, marshy deposits or oncolytic sands layer are interstratified within the anthropogenic layers of the first levels of occupation on several cores (Lespez et al., 2013). It indicates a rise of the water table of the little pond located at the bottom of the site which is feed by an exsurgence in the marble slopes which dominate the site (fig09). On C3, it corresponds to 2 high-stands. The first one is dated on C3 after 8.38/8.17 ka while the second is dated on C2 and C3 around 8.0-7.9 ka. Additionally, the geomorphological observations in the Dikili Tash small valley which runs to the marsh show development of detrital carbonate sedimentation. On Dik4 core, it correspond to a carbonate silty layers which interrupted the organic sedimentation. It suggests an increase of flood flows from the small stream which runs from the Dikili Tash pond during the period 8.2-7.8 ka. These observations are close to the results obtained at Lake Doirani (130 km WNW) (fig02) which show a relatively high lake level during this period (Zhang et al., 2014). From the beginning of the 9th millennium cal BP, the vegetation cover shows the return of some pioneers or mesophilous taxa (hazel, elderberry and black haw trees), or their appearance (ash and broom) shortly before a closing landscape phase. Locally, the riparian vegetation increases considerably in relation to a drier



environment due to previous detritic sedimentation input which fill the edge of the marsh and the
water level decrease begun from ca 7.5 ka. and the forest cover expanse more generally in the region
in relation with climatic amelioration (Pross et al., 2009).
3.4. NW Grece-Corfu Island
The prehistoric site of Sidari, located in a small coastal valley dug in marine Pliocene detrital
formations of NW Corfu Island (fig02 and 10a), is a crucial milestone to explain the modalities of the
Neolithisation phase in the Adriatic zone. It is the oldest Neolithic Site of Central Mediterranean (8.3
ka) (Berger et al. 2014). This coastal sector is part of a vast tertiary sedimentary basin with a hilly
morphology that displays vast and deep Holocene alluvial formations. Rainfall is today extremely
important with an average of 1000mm/year, in the most humid region in Eurasia at this latitude (39°
N). This situation is explained mainly by its location close to the Balkan mountain barrier to the east
of the Adriatic zone. Both valleys studied (Sid. 1 and 2) are tributaries of the small coastal Peroulades
River (fig010b), providing sustainable water resources, a rich wetland habitat and deep alluvial soils to
its occupants.
The geoarchaeological study compares two lower rank watersheds close to 400m. The outdoor
stratified archaeological site (Sid.1) and the neighbouring small valley (Sid.2), both located in the
valley floor present a strong dilatation of the sedimentary sequence (5 to 7m), a succession of
Holocene paleosols and a highly favourable hydromorphological context (interlocking channels).
Sidari 1 is associated with a dense archaeological occupation and Sidari 2 with a much less
anthropised and deeper archive (fig04a). A precise field geomorphological and palaeopedological
approach, favoured by the presence of interbedded archaeological levels and charcoal beds which are
systematically radiocarbon dated, allowed the construction of a solid micro-regional
chronostratigraphic framework. A CPDF analysis is used to better specify the chronology of
hydrosedimentary and pedological activity. A local database integrating Sidari 1 and 2 sites was
compiled. It integrates 33 radiocarbon dates from 3 main contexts: channel fillings, floodplain
overbank deposits and palaeosols. They were generated using the guidelines set out by Johnstone et al.
(2006). BP calendar ages, including 1s error range, were summed using a macro excel software. This
analysis provides a probabilistic assessment of centennial-length sedimentary aggradationnal episodes
interrupting Early Holocene active pedogenic and landscape stability development favoured by a more
humid Mediterranean climate within 2 individual catchments.
Sid.1 archive presents a 5m pedosedimentary sequence depth. The rescue archaeological excavation
operated in the mid-2000s had uncovered 8 main archaeological layers from the Mesolithic to the
Helladic periods that are interbedded in a complex polyphased sequence, with 16 main phases of river
and colluvial activity and pedogenesis in 5 millennia (Berger et al. 2014) (fig10c). Sidari 2 is a natural
transversal trench of a small dry valley, 80m wide and 7m deep, entrenched in cemented Pleistocene
formations. The deposits are actively eroded by the current sea level change that allows a full
observation of the Holocene filling to be performed. A first chronostratigraphical view of the sequence
identified 2 abrupt limits at the Early-Mid Holocene (around 8.2 ka) and Mid-Late Holocene periods
(around 4.0 ka) (fig10d) which refer to the recent tripartition of Holocene period (Wanner et al. 2008).
In this paper we focus only on the lower half of the filling, consisting of a thick cumulic soil complex
and the beginning of the mid-Holocene period marked by a a rapid breakdown of pedosedimentary
conditions, driving to a very erosive and detrital activity in the small marly basins during 1
millennium.
The Sid.2 local chronostratigraphy building clearly presents a stairway age depth model with three
phases of high acceleration of sedimentation rate (fig11b): from 10.4 to 10.0 ka, from 9.5 to 9.0 ka and
after 8.4 ka. This environmental temporality clearly represents millennial pedogenesis/incision-
aggradation rythmicities, particularly well illustrated in the Sid.2D profile (fig11a) which represents a
morphopedological synthesis of the events succession. A systematic sedimentological and
geochemical multi-proxy approach that describes pedoclimatic conditions, hydrosedimentary
environments, detrital fluxes and some ecological factors (fires) is still forthcoming.
Hydrosedimentary and paleopedological interpretations presented in this paper should be viewed as
preliminary.



The biostability phases that develop between erosive phases discussed are expressed in geological records of catchment heads by a black deep soil development (phases I, III et V, fig11a), often decarbonated and leached, as observed at the microscopic scale in Sid.1 (Berger et al. 2014). These kinds of pedogenesis and associated pedofeatures (hyaline cutans) illustrate a dense forest cover highly protective for soils (Macphail et al. 1987, Kühn 2003). Local charcoal assemblages (Delhon and Thiebault forthcoming) and the regional pollen spectra (Bordon et al. 2009, Triantaphyllou et al. 2009, Combourieu-Nebout et al. 2013, Glais et al., 2016) reveal vegetation dominated by mesophile deciduous oakforest. Following a first broadly stable and humid Holocene, favourable to the development of a thick leached and humic cumulic palaeosol (Berger et al. 2014), the second half of the Early Holocene is punctuated by a succession of abrupt breaks in the hydromorphological functioning of the marly valleys, of centuries-terms, and of quasi-millennial cyclicity. They are characterised in the field by a sudden stop of soil formation processes, synchronous of deep gullies which fit into each other during the three EH climate events (fig11a). These gully activities (phases II, IV, VI) are followed by a rapid-filling phase of lighter tone alluvio-colluviation often still decarbonated (association of inherited soil material and marls) which palaeodynamic can be characterised by analysis of the sedimentary fill mode: (1) The slick or lenses sand and gravel deposits, rich in small well-rounded nodules of clay soil are associated with concentrated runoff causing gullying and sapping upstream soil formations (fig11cd-IVb1-VIb) and (2) finer well-sorted deposits, often micro-laminated, associated with finer and regular rainfall generating diffuse runoff (fig11e-VIg). So we explain the formation of these two facies by the expression of different rainfall on largely bare surfaces by fire (regular charcoal beds presence). The transition between RCC events and the pedological stabilisation of the valley is generally dominated by more regular rainfall (fine granularity, diffuse laminations), as in the 8.2 ka event.

The 10.4-9.75, 9.5-9.1 and 8.35-7.9 ka active periods are individualised using cumulative probability density functions (CPDF) plots (fig11f). We interpret these morphological and hydrosedimentary signatures, regularly recorded in alluvio-colluvial archives at Sidari, as the manifestation of rapid climatic changes (RCC), which seem to form the rhythm of the evolution of Holocene north Mediterranean valleys.

It especially allows hypotheses to be proposed about the potential climate impacts on continental hydrology, soils, and vegetation dynamics in relation to the development of human societies on the micro-regional level. These new data establish the necessity of always reasoning from contextualised data, not to be taken hostage by temporal CPDF-type constructions, sometimes too schematic and occasionally disrupted by bias related to the organic material used for the $^{14}$C. Indeed, we observe a constant time lag between chronocultural and morphological data (from 100/150 yrs) whose origin is probably to be found in the old wood effect (almost a predominance of oak in charcoal assemblages). The Sid.2 Mesolithic occupation centred on the 9.3 ka event is associated with a short intermediate RCC pedogenic episode. The new Sidari chronostratigraphical context does not identify one Mesolithic horizon, but probably 3 successive ones. Cultural continuity proposed by Sordinas (1969, 2003) is only apparent, as produced by geomorphological impacts of the 8.2 ka event (Berger et al. 2014). The Early Neolithic I "monochrome" occupation sets up on the paleosol (S3) before being partially eroded (fig11e), and the last diffuse occupation levels of EN.I then interbedded in the first aggradation levels of the 8.2 ka event (AP5). Finally, the Early "Impressa" Neolithic II level is clearly associated with the intra-8.2 ka episode of soil stabilisation in SID-1, then covered by the second stage of alluvial aggradation (AP6). If we think in radiocarbon time, the gap initially mentioned by Sordinas (1969) between the two horizons of Early Neolithic (Monochrome and Impressa) is very brief (a few decades at most). It is much more marked in the sedimentary archives studied, as amplified by the very rapid aggradation process of the 8.2 event. This second peak of 8.2 hydrosedimentary activity (AP6) seems to correspond to a durable site abandonment (until Late Neolithic) (cf. Berger et al. 2014).

**4. Discussion about Early to Mid-Holocene RCC impacts on terrestrial hydrosystems and human societies at the North-Eastern Mediterranean scale**

The results obtained on the 4 sites studied assess the local environmental changes which can be linked





to the RCC changes. In particular, they underline the sensitivity of hydrosystems and vegetation to
climatic changes at a secular scale. We show that the SH cooling event, correlated with glacial
outburst in the Northern Atlantic, low values of total solar irradiance and K+ records in Greenland ice
cores, have a major impact on the functioning of central to eastern Mediterranean continental
hydrosystems (fig11a).
The 9.2 ka event matches one of the largest early Holocene meltwater pulses at 9.17 ± 0.11 ka B.P.
(Teller et al. 2002) which was probably triggered by a slowdown of thermohaline circulation. In the
Asian monsoon domain (Qunf and Dongge caves) stalagmites shows a positive anomaly in $d^{18}O$
calcite at 9.2 ka reflecting lower monsoon precipitation (fig01). The duration of the event is less than
150-200 years in all records discussed by Fleitmann et al. (2008). A recent metadata analysis of
Holocene European river activity highlights the current lack of well-dated records for the Early
Holocene with only two Iberian flood clusters (9.5–9.2 and ca 9.0-8.8 ka : Benito et al. 2015), in-phase
with high lake levels in the Jura Mountains and the northern French Pre-Alps (9.55-9.15 ka : Magny,
2004). Both records likely reflect their high sensitivity to North Atlantic circulation. In Sidari 2 valley,
a large signal of gully erosion and vertical aggradation is synchronous to the European lakes and
Iberian rivers record, with two activity peaks between 9.5 and 9.1 ka (fig01). Comparable signals
before 9.0/8.9 ka do not occur in the hydrosystems of the central Anatolian plateaus (fig04) which
respond to a high humidity in the Taurus range that feeds the high water levels in lakes and marshes
located at the foot of the Taurus. But the strong drying signal from 9.0/8.9 ka is well registered by the
hydrosystems in Sidari and central Anatolia as well as by the vegetation cover on the Aegean and SE
Balkans areas.
The 8.2 ka Hudson event is recorded, at all the sites presented here. In the area, it occurs during a long
cool interval beginning ca 8.6 ka (Rohling, Pälike 2005). Like the Northern Aegean and Ionian
terrestrial archives discussed by Weninger et al. (2014) and Flohr et al. (2015), we discuss below the
bi-partition of the event in an earlier phase (a cold phase from 8.5-8.4 to 8.2 ka amplified during a later
phase (a RCC 8.2-8.05 ka) by the Hudson Bay outburst, followed by a third sub-phase between 8.05-
7.9 ka in the northern Greek and central Anatolian archives that we call C (fig12a).
4.1. Sidari/Dikili Tash and the EH northern Greece/southern Balkan regional pattern
The increase of erosion and fluvial activity observed on both archaeological sites around 8.2 ka has
also been observed elsewhere in northern Greece as in the Lake Prespa (Panagiotopoulos et al., 2013)
and Lake Doirani (Zhang et al., 2014) (fig02, 12a) areas. It confirms the trends of increase of soil
erosion and sediment transfer to the wetland around 40-41° N during the 8.2 ka event. At the regional
scale, these continental results seem consistent with the Adriatic climate data from NW Greece to the
Po Valley in northern Italy. The confrontation with the nearest multi-proxy marine records (MD 90-
917 in the central Adriatic sea) and northern Aegean Sea consolidates the regional climate-
environmental mechanisms previously described (Rohling et al., 2002 ; Khotthoff et al. 2008,
Combourieu-Nebout et al., 2013 ; Berger et al., 2014) (fig02,12a). The pollen of deciduous oak forests
(reflecting tree cover peri-Adriatic mountain) sharply decrease to each hydromorphological failover
observed in Sidari, in synchrony with the RCC, around 10.1, 9.2 et 8.3 ka (Combourieu-Nebout et al.
2013). This functioning coincides with the dominance of coniferous forest (mainly firs) at high
altitudes at ca. 8.5-7.8 ka (Lakes Ribno and Trilistnika, southwestern Bulgaria) (Tonkov et al. 2013)
and with the replacement of *Quercus* dominated forests with mixed deciduous forests at around 8.3 ka.
These regional evolutions underline the role of climate change and cooling more than the
consequences of the onset of agropastoral activities during this period.
Nevertheless, the observations made on the edge of the Tenaghi-Philippon marsh evoke questions. In
fact, from 8.4 to 8.1 ka, a general cooling has been recorded by recent Holocene palaeoclimatic studies
in the Tenaghi-Philippon marsh (Pross et al., 2009) and northern marine Aegean region (Kotthoff et
al., 2008) with an interruption in Sapropele 1 formation (fig02). They propose a scenario of
deteriorated winter climate conditions with temperatures lowered by more than 4°C in winter, less
than 2°C in summer (Pross et al., 2009). Sea surface temperature from the core MD 90-917 in the
central Adriatic Sea (fig02) also indicates a decrease of at least 2° C between 8.3-8.1 ka (Combourieu-
Nebout et al. 2013). Davies et al. (2003) identified a strong decrease of summer temperatures at the
same time at the scale of Southern Europe (8.3-7.8 ka). This is explained by an increase of outbreaks



of cold and dry air from higher latitude (Siberian high) (Rohling et al. 2002; Marrino et al., 2009). The climate was drier and characterised by a decrease of annual rainfall by 800 to 600mm due mainly to a decrease of winter precipitation. To explain the apparent contradiction between the local pollen and geomorphological data and the regional climate reconstruction from pollen data, we suggest that the cooling was favourable to the development of snow cover and associated spring-flood flows and to reduction of evapotranspiration (Lespez et al., 2013) or Tenaghi Philipon sampling is not precise enough to describe the internal structure and moister episodes of the 8.2 ka event. Moreover, it appears that the summer rains increased during this period (Peyron et al., 2011) limiting summer evapotranspiration and probably the decrease of the water table as observed for Late Quaternary cold periods in Anatolia for example (Jones et al., 2007). Thus local water balance can be different of the regional trend which is, moreover, not indicative of the flood flows energy and frequency. It appears that cold air SH extension mixed with the warmer air over the Mediterranean, may have created a surplus of potential energy resulting in regional cyclogenesis (Makorgiannis et al., 1981) from spring to fall triggering significant flood flows in the studied areas. Increase of climatic instability and summer rains may explain the hydrogeomorphological signals of Sidari 1 and 2 valleys. The repeated succession of gullies and torrential discharges between 8.4 and 7.9 ka (fig11acd, 12a) could be associated with concentrated summer rains and increase of climatic instability.

## 4.2. The potential impact of the 8.2 event on Societies

### 4.2.1. An impact primarily focused on readability of archaeological archives

Firstly, our data show that the 8.2 ka event played a significant role in the archaeological records. Indeed truncature and hiatuses correspond to erosional events or riverscape changes more than abandonment of inhabited areas. It explains, for example, the archeological continuity which led the first archaeologists of the site to suggest the hypothesis of a "Sidarian" Neolithic inherited from an existing local Mesolithic. Alluvial truncations moved sedimentary horizons of these 2 cultural periods (by sediment ablation) and may even have associated them within alluvial formations where we found reworked Mesolithic and early Neolithic material and charcoal (Berger et al. 2014). New data and reinterpretation of old archaeological data illustrate a strong erosion phase at the Mesolithic-Early Neolithic transition in the Central Mediterranean area (Mlekuz et al. 2008, Berger et Guilaine 2009, Berger et al. 2014). A similar process is observed in the Eastern Mediterranean area in the Khirokitia sites (Cyprus) where at least 2 episodes of fluvial discharges, flash flood types, strongly impact the Neolithic village (Hourani this paper). The same dynamic is observed in Ain Ghazal, Wadi Shu'eib and Abu Thawwab in the Levant where densely-packed layers of cobble deposits are observed between late PPNB and PN archaeological horizons (Simmons and Mandel 1988), with a permanent uncertainty about the absolute chronology of these events after the remobilisation of [14]C dated old bones (Zielhoffer et al. 2012). Even in protected contexts such as Western Albanian mountains caves in front of Corfu Island, geoarchaeological research identified a long slope instability period responsible for a partial erosion of the archaeological deposits (8.2 event effects?) (Schuldenrein 1998) (fig10a, 13) synchronous of Sidari valleys geomorphic changes. In some floodplains, even if the fluvial activity did not imply high energy event, as in Dikili Tash, the increase of water level may change the location of the inhabited areas. There, the vegetation cover and hydrosedimentary changes were the result of change in climatic conditions and the development of anthropisation. The marshy and fluvial sedimentation interrupts the archaeological sedimentation on C3 and C2 and reaches 53-54m above sea level. However C10 and C1 located slightly higher on the former alluvial fan, 54m above sea level, show the continuation of the settlement during the 8.15-7.8 ka period (fig9, 13). So, it is noticeable that the climatic change and its geomorphological consequences do not infer a notable hiatus in human occupation, but probably merely a local displacement and relocation of the settlement on the tell (Lespez et al., 2013, in press). At the same time deep explorations of Macedonian floodplains attest to the presence of Neolithic levels under several metres of alluvial sediments (Lespez et al. 2014), that raise questions about the extent of the still-hidden archaeological reserve. Obviously the few examples discussed clearly illustrate that 8.2 event geomorphological evolution plays a major role in the distortion of the first Neolithic signal, in the NE to Central Mediterranean zone where Neolithisation occurs and advances just before the 8.2. event. The strong rainfall



irregularity that seems to characterise the period around the 8.2 event, could be the cause of these
repeated impacts on Neolithic river sites (fig13). The greatest contribution of the summer rains
(Peyron et al. 2013) may be an explanation for the observed hydrogeomorphological functioning
between Cyprus and the Balkans and the difficulty to link environmental changes and settlement
history as in the 3 sites evoked.
4.2.2. 8.2 Event and "Neolithic go to West" onset?
The question that now arises is, in the case of western Anatolia, why and how the diffusion of
Neolithic practices occurred from the central plateaus towards the Aegean region, and at what such a
speed. The Early Neolithic is rooted in local PPN cultures at Catalhöyük-East ca 9.4/9.3 ka cal BP, in
the Lake District ca 9.2/9.1 ka cal BP, and possibly in the Aegean region (Ulucak) around 9.1 ka cal
BP (fig14). In these specifically local contexts, pottery appears about the same timing in all excavated
sites between 9.0 and 8.8 ka cal BP. Would the answer to the "why" be related to the appearance of
pottery, which may have increased the capacity of humans and animals to travel and start to develop
contacts on the regional scales?
From 8.6 to 8.0 ka, the cultures of Yarmoukian (Southern Levant), Khirokitian (Cyprus),
Monochrome (Western Anatolia, Aegean world) (fig02) are directly confronted by the climate change.
There is also manifold evidence for population movements in coastal and low-lying locations in the
Northern and Southern Levant, and finally with the abrupt appearance of Neolithic communities in the
Aegean/Ionian zone, where Dikili Tash and Sidari are located (Weninger et al. 2014). Weninger et al.
(2006, 2014) suggest that climate-induced crises may have forced early farming communities to
fission and move in order to escape new conditions and possible related conflicts (scalar stress). In the
first phase of the 8.2 RCC (8.6-8.3 ka : phase A), there is evidence of a push/pull to coastal and lower-
lying locations in the Southern Levant and Anatolia after Clare (2013), but this trend hypothesis seems
questionable from Flohr et al. (2015) and from the anatolian data discussed in this paper. As coastal
and lower-lying areas would have been less affected by typical RCC-impacts (drought and severe
winters) (Weninger et al. 2014), the related abandonment of sites in Jordan, in the northern Levant,
Eastern Anatolia and Cyprus is referred to as 'Late Yarmoukian Crisis'. This cultural event coincides
for the authors with a further wave of Neolithic expansion into Southeast Europe in the second phase
of RCC (8.3-8.0 ka : phase B). But in the light of 3 new radiocarbon data series (with charcoals and
shortlived species) on the early Neolithic from northern Greece and of new clear geoarchaeological
contexts, we propose a different temporal timing for Northern Greece colonisation than Weninger et
al. (2014) by demonstrating the anteriority of Neolithic migration from western Anatolia (Dikili Tash,
Sidari, Mavropigi-Filotsairi and Nea Nikomedia) to the second phase (B) of 8.2 ka events, sometimes
far to the West. This assertion is also based on local chronostratigraphic and geomorphic contextes in
Sidari and Dikili Tash, which illustrate the posteriority of hydrogeomorphological and erosion
signatures to Neolithic implantations (fig12a, 13). The chronology of this northern Greece Neolithic
package implantation would no longer be synchronous with the strictly speaking 8.2 event (glacial
outburst derived effects), whose minimum time is estimated between 8.2 and 8.05 ka in the more
precise glacial and speleothem proxy data (fig01) but could be in adequation with the more general
aridification/cooling from 8.6/8.5 to 8.0 (Rohling and Pälike 2005, Göktürk et al. 2011). The earliest
spread of Neolithic packages to Western and Northwestern Anatolia occurred almost a thousand years
before the 8.2 ka event as illustrated by recently-published robust chronological studies (Özdogan et
al., 2012a, 2012b; Düring, 2013; Clare 2013, Brami, 2014, Kuzucuoglu, 2014; Stiner et al., 2014;
Weninger et al. 2014, Flohr et al. 2015) (fig12a, fig.14). The question that now arises is whether the
diffusion of Neolithic practices which began in the Central Anatolian highlands around 8.7 ka did not
include at the same time and in a same cultural stream the northern Aegean area to the southern
Balkan borders (Thracia, Macedonia, Thessalia), but by taking the recent pattern of Weninger et al.
(2014) from the middle of phase A (fig01, 12b) and not during phase B, in a rapid colonisation
movement that fits in continuity from the highlands of central Anatolia (median speed of Neolithic
wave of advance from 4 to 6 km/yr). We have not to forget in the general Neolithic mobility trend
from Anatolia that Franchthi cave (Argolid) was occupied by farmers around 8.6 ka (new dates on
seeds) (Perlès et al. 2013), not much later than the earliest occupation of Knossos in Crete (Efstratiou
et al. 2004). These data out of doubt support a southern route and a model of multiple origins for the



introduction of the Neolithic in Europe. To temporally have hemispheric aridification identified in the
various marine and terrestrial climate-environmental proxies coincide with the Neolithic population
movement from Central Anatolia, should be according to the latest CPDF treatments proposed by
Flohr et al. (2015) that aridification begins at least at 8.7 ka (by reasoning with either the total
radiocarbon or "shortlived" dates available for western Anatolia. However, the overview of the current
multi-Proxies data identifies a real general trend from 8.6-8.5 ka (fig12b) and real continental
hydrogeomorphological evolutions seem to occur only from 8.4 ka (fig13). Can this lag be attributed
to the age models used in the environmental series? The reservoir effects cannot be challenged here
since western Anatolia chronocultural series are based on a robust set of shortlived dates. Furthermore,
the results obtained in central Anatolia underline the contrasted response, in time and in space, of the
local environment to RCC (fig14). Alluvial fans of the Taurus piemonts stops to aggrade from 8.5 ka
to 8.0 and paleosols are recorded between 8.2/8.1 and 8.0/7.9 ka, illustrating a dryer period which
seems to have begun earlier in other Central Anatolian highlands (Bor Plain, Tuz Gölü, Akgöl marsh)
around 8.9 ka and, in the Bor plain, a humid period is recorded from 8.5 to 8.1 ka, before a fast, sharp
drop in the aquifer. The hypothesis of a trigger foremost cultural shall also be considered ; the ball is
now in the culturalists camp.
The second "European" step took Neolithic lifestyles away from the Aegean coastline all the way to
continental Bulgaria and Serbia by the main river axis (Struma, Vardar, Maritsa) and could be
associated to the Dʃuljunica (Raiko Krauß et al. 2014), Anzabegovo (Gimbutas 1976) and Kovacevo
(Lichardus-Itten in press) pre-Karanovo sites just after the Hudson Bay event (around 8.1 ka), i.e.
almost 200/250 years after the first European Neolithic wave. We must now integrate into the coming
socioenvironmental discussions on the steps of the Neolithic diffusion through the Balkans and the
Adriatic a last shudder of 8.2 event between 8.05 and 7.9 ka (fig01-green, i.e Lake Maliq, Qunf cave,
Sofular, Steregiou, marine core SL 21, Sidari). Episode clearly in step with a peak of [K +] on GISP2
and a small bond event. We enter here in a temporality of 8.2 event that was little discussed, that of a
possible tripartition of the event we are trying to argue based on Sidari (Berger et al. in progress) and
Dikili Tash records. A two-stage cooling around the time of the 8.2 ka event has been identified in
speleothems of Ireland (Baldini et al., 2002), in pollen diagrams from Central Europe (Lotter and
Tinner, 2001), lacustrine records in Norway (Nesje and Dahl, 2001), and a two-step release of Lake
Agassiz waters has been modelled by Clarke et al. (2004). The marine data of LC 21, SL21E and
MD952043 also show two colder peaks separated by a temperate rise, while Dongge Cave $\delta^{18}O$
(Wang et al., 2005) and Qunf cave isotopic data illustrate two hyper-arid episodes separated by a
wetter episode. We find here the most complex structure of the 8.2 event discussed by Thomas et al.
(2007) based on isotopic data of GISP2 et GRIP. We must now integrate this new climatic and
environmental temporality to the classical Neolithic wave of advance hypothesis, if they are linked.
The challenge is open.
4.2.3. Climatic event and social impact
More fundamentally, the impacts of climatic changes or natural extreme events have to be evaluated in
terms of biophysical and social vulnerabilities. Burton et al. (1993, p35) refer to the seven dimensions
of hazardous events: magnitude, frequency, duration, speed of onset, geographical extent, spatial
dispersion, and temporal spacing. However, as underlined by Clare and Weninger (2008), impacts
upon the resources of a society are primordial (availability of natural resources), and responses in
terms of resources addressed (variety), of land use (management), technology (tool production,
equipment progress, variety), housing quality and residence location adaptability have to be
considered. Social vulnerability studies must consider the societal perception of the causes of
environmental change (Blaikie et al. 1994) and the efficiency of social communication processing
(Van der Leeuw et al. 2009). There is also a need for more site-specific detailed studies focusing on
ecological bases and strategies (Flohr et al. 2015). Only such new trajectories, closely interlinked with
the intra-archaeological sites multidisciplinary analyses will optimise our perception of forms of
socioenvironmental resilience. Concretely, for the period and the studied areas, the abrupt global cold
events might have affected the vegetative season time, growth of wild plants and predictability of food
resources. Loss of soil cover potential (by erosion), dryness or wetness effects on soil productivity
could be directly or indirectly documented by quantitative climate reconstructions from pollen





diagrams (Peyron et al. 2011) to discussion of the agrarian constraints during RCC events. Recent fire
signal studies in eastern Mediterranean (Van Lake (Wick et al. 2003), Dikili Tash, this study, Sidari in
progress) document dryness, available fuel and variations in vegetation cover and have to be
systematised in future research to better discuss their link with climate changes and human impact on
vegetation. Nevertheless, we must keep in mind that the geographical setting of the eastern
Mediterranean results in physically very contrasting environments in which it is often sufficient to
move over very short distances to find different environmental conditions (Willcox, 2005; Lespez et
al., accepted). In fact, a dry period could imply a move closer to water resources or, on the contrary, as
observed in Dikili Tash, a rise of water table and flood hazards might imply leaving the floodplain to
settle higher on the alluvial fans or lower slopes in the surrounding areas. The uneven exploratory and
excavation practices on sites and around sites are to question: the lack of extensive archaeological
excavations on most reference Neolithic sites (and our uncomplete knowledge of the other ones too,
an information crudely lacking when discussing occupation dates and periods) strongly hampers
interpretations on the continuity of Neolithic occupations and therefore and does not always decide on
climate impacts on societies. Furthermore, Neolithic communities rely on diverse subsistence
strategies including wild resources (Asouti and Fuller, 2013) even during more recent periods
(Valamoti, 2015). Finally, the resilience of the early farming societies should not be underestimated
(Flohr et al., 2015).
**Conclusion**
This study demonstrates the reality of hydrogeomorphological responses to early Holocene RCCs
derived from glacial outburst in valleys and alluvial fans and lake-marsh systems. It highlights the
importance of Holocene sedimentation and post-depositional disturbances on reading the Mesolithic-
Early Neolithic transition and attestation of the first true levels of Neolithic occupation in South East
Europe. Terrestrial records still reflect heterogeneities in paleoclimatic restitution across the north-
eastern Mediterranean during RCC events (from central Anatolia to southern Balkans). This signal
heterogeneity shall now be discussed in terms of quality of exploited archives, of sampling/measuring
time resolution and of regional climatic pattern variations. The widespread use of Core scanner
geochemical analysis will promote the identification of the finest Holocene variations. The issues are
important to better assess climate impact on the functioning of coastal and continental environments,
in major societal disruptions such as the Neolithisation of the Mediterranean. Research on the effects
and impacts of 10.2 and 9.2 RCCs are still in their infancy. They are potentially present in continental
sedimentary archives and shall be better understood in a socio-environmental perspective. Our
hypothesis of an early Neolithic colonisation of the North Aegean (around 8.4 ka), prior to the
assertion of the second and more marked part of the 8.2 RCC event should be supported by new data
in the coming years thanks to the increasing number of deep trenches and core drilling in regional
river and marshy areas, including the immediate vicinity of the main Neolithic tells whose first
sedimentary archives are still often unknown. The simultaneous achievement of pollen studies with
very high time resolution will complete the approach to attest to early agricultural practices. These
data must be compared to precise archaeological data in order to assess the impact of the climatic
changes on the environment and the farming societies at the local scale. Rather than collecting
radiocarbon dates in order to propose modelisation of Neolithic expansion, we need to have more case
studies at the regional and the Eastern Mediterranean scale if we want to discuss reasonably the role of
climatic changes in cultural transformation.
Aknowledgments :
We thank Paléomex-Archéomède (CNRS INEE) for their financial and logistic support. This paper
has benefited from the financial support from the ENVOL program of the Mistral project. Many
thanks to the french missions of Foreign Office (MAE) in Greece (Dikili Tash, Darcque, P. and
Tsirtsoni, Z.) and in Cyprus (Khirokitia, A. Le Brun). We don't forget the Collège de France, « chaire
des Civilisations de l'Europe au Néolithique et à l'âge du Bronze » and The 8th Ephoria of prehistoric
and historic antiquities of Corfou (G.Metallinou).We finally thank the Laboratoire de Mesure du
Carbone 14, UMS 2572, ARTEMIS in Saclay for [14]C measurements by SMA in the frame of the





National Service to CEA, CNRS, IRD, IRSN et Ministère de la Culture et de la Communication.

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

**Figures**
Fig.1. Northern Hemisphere Palaeoclimate/pedosedimentary records illustrating Holocene Rapid
Climate Changes (RCCs); 1. Greenland GRIP ice-core δ18 O (Grootes  et al. 1993 ); 2. High-
Resolution GISP2 nss [K+] as proxy for the Siberian High (Mayewski  et al. 1997), 3. Ice rafted debris
in Northern Atlantic (Bond et al.. 2001), 4.  Eastern Aegean SL21 (Sea Surface Temperature, SST)
fauna (Marino  et al. 2009 ), 5. MD952043 SST, 6. (C) Eastern Mediterranean core LC21 (Sea Surface
Temperature, SST) fauna (Rohling et al. 2002 ); 7. Steregiou (Romania) Pollen-based temperature of
peat pollen (Feurdean  et al. 2008), 8. Sufular Cave δ$^{13}$C (Northern Turkey, Fleitmann  et al. 2009 ), 9.
Lake Maliq Pollen-based temperature of the coldest month (Bordon et al. 2009), 10. Qunf cave-Q5,
$^{18}$O (%0 VPDB) (Fleitmann et al. 2003), 11. Sidari valleys 1 and 2 (Corfu island) Gully erosion/fluvio-
colluvial aggradation (CPDF this study), 12. Sidari valleys 1 and 2 Soil formation phases (Corfu
island) (this study), 13. Tenaghi-Philippon N-Greece Tree-Pollen (%)(Pross et al. 2009). Yellow
vertical bars underline the 9.3 and 8.2 ka events phases. The yellow, orange and green bars (associated
to A, B, C letters) represent a possible tripartite temporal structure of the 8.2 ka event (discussed in the
text).
Fig.2. Map of main sites cited in the text. 1.Lake Accesa, 2.CM-92-43, 3.MD90-917, 4.MD 04-2797,
5.Dfluljunica, 6.Anzabegovo, 7.Lake Trilistnika, 8.Lake Ribno, 9.Kovacevo, 10.Lake Dojran, 11.Dikili
Tash, 12. Tenaghi-Philippon marsh, 13.Nea Nikomedia, 14.Mavropigi-Filotsairi, 15.Paliambela,
16.Lake Prespa, 17. Lake Maliq, 18.Sidari 1/2, 19.Konispol cave, 20.SL-152, 21.KL-71, 22. Ulucak,
23. Sofular cave, 24. NS-14, 25.LC-21, 26.Hacilar, 27.Lake Golishar, 28.Çatalhöyük, 29.Can Hasan,
30.Musular, 31.Aşıklı, 32.Khirokitia-Maroni River, 33.Vasilikos Valley, 34.Shillourokambos, 35.Tell
Sabi Abyad , 36.Soreq cave, 37.Wadi Shu'eib , 38.Ain Ghazal, 39.Dead sea., 40. Franchthi cave, 41.
Knossos. Main Neolithic cultures of the 9th millenium cal. BP are in blue.
Fig. 3. The main large plains of endorheic central Anatolia and location of sites cited in the text and in
fig.7. Main cities: K: Konya; E: Ereğli; B: Bor; A: Aksaray. Palaeoenvironmental sites: 1: Yarma
(Kuzucuoğlu et al., 1999); 2: Çarsamba fan (Boyer et al., 2006); 3: Sultaniye (Kuzucuoğlu et al.,
1997); 4: Karapınar sand dunes (Kuzucuoğlu et al., 1998); 5: Düden (Fontugne et al., 1999;
Kuzucuoğlu et al., 1999); 6: Adabağ (Bottema and Woldring, 1984); 7: Zengen; 8: Bayat; 9: Kayı
(KKK); 10: Pınarbaşı; 11: Bahçeli; 12: Sazlıca; 13: Melendiz-Çiftlik (Kuzucuoğlu et al., 1993); 14:
Alluvial fans (Naruse et al., 1997; Kashima et al., 2002). Sources for 7 to 12: Kuzucuoğlu et al., in
prep. Excavated Neolithic sites cited in text: a: Boncuklu; b: Aşıklı; c: Can Hasan III; d: Çatalhöyük
East; e: Tepecik-Çiftlik; f: Pınarbaşı-Karadağ; g: Pınarbaşı-Bor; h: Köşk Höyük; i: Çatalhöyük West.
Fig.4. Dated palaeoenvironmental records in the three main endorheic plains of central Anatolia: a
synthesis between 12.5 to 6.0 ka cal BP. Environmental records in sediment archives: 1. Deep lake; 2.
Backswamps; 3. Vegetated shallow marshes; 4. Palaeosol; 5. Alluvial fan (coarse sediment). Humidity
intensity (synthesis): 6. Dry to very dry; 7. Emersion of watered ecosystems and soil formation; 8.



Semi-arid and/or contrasted seasonal climate (high seasonal run-off); 9. Humid (marshes); 10. Very humid (lakes, backswamps).

Fig. 5. A/General location of the Pre-Pottery Neolithic site of Khiroktia and of the Vasilikos Valley, mentioned in the text ; B/Topographical map of Khirokitia illustrating the position of the site comparing to the River Maroni and the location of the different studied areas.

Fig.6. A/Synthetic cross section of the Maroni Valley at the foot of the eastern slope of the site showing the depositional environments of the river and the situation of the studied archaeological sequence. The location of the section is shown in figure 5b ; B/North-South section through the occupation levels at the river border (operation 2) with the stratigraphic position of the major erosional event 2.

Fig.7. The Tenaghi-Philippon (former) marsh, Dikili Tash archaeological sites and sample cores obtained from the marsh deposits mentioned in the references. Image from Google Earth (40°58'0N, 24°15'0E).

Fig.8. Diagram from the Dik4 core. LOI and Carbonate content of the sediment expressed in % of the total sediment. Charcoal influx expressed $cm^{-2}.yr^{-1}$. Selected pollen and NPP groups expressed in % (see Glais et al. 2016) : 1) xerothermophilous taxa (*Ephedra fragilis* type, *Erica arborea* type); 2) ruderal taxa (*Asphodelus albus* type, *Asphodelus fistolosus* type, *Boraginaceae*, *Cannabis/humulus* type, *Cardueae*, *Centaurea nigra* type, *Fumaria officinalis*, *Malva sylvestris* type, *Rubiaceae, Rumex acetosa* type); 3) anthropozoogenous taxa (*Plantago lanceolata* type, *Plantago coronopus* type, *Polygonum aviculare* type, *Urtica dioica* type, *Vicia* type); perennial pasture plants (*Apiaceae, Brassicaceae, Caryophyllaceae, Fabaceae* undiff, *Gentianella campestris* type, *Helleborus foetidus* type, *Jasione* type, *Primulaceae*); coprophilous, NPPs (*Cercophora* sp. Type 112, *Podospora* sp type 368, *Sordaria* sp. Type 55A, *Sporormiella* sp. type 113, *Coniochaeta cf. lignaria* type, *Ustulina deusta* Type 44); eu-mesotrophic NPPs (*Ceratophyllum* sp. Type 137, *Botryococcus* Type, *Gloetrichia* type 146, *Spirogyra* Type, *Neorhabdocoela* undiff., Type 128A, Type 18 Type 151, *Zygnema* Type); meso-oligotrophic NPPs (*Anabaena* sp. Type 601, *Rivularia* Type 170); NPPs indicative of erosive processes (*Glomus cf. fascilicatum* type 207 and *Pseudoschizaea circula* type); NPPs indicative of fire events or dry conditions (*Chaetonium* sp Type 7A, *Neurospora* sp. Type 55c, *Pleospora* sp. Type 3B, Type 200).

Fig.9. Map of the core drillings around Dikili Tash site and interpretation of the settlement dynamics during the early stages of the Neolithic.

Fig.10. A/Map of the Corfu island with location of the site of Sidari on the northern coast, B/ Location of the Sidari 1 and 2 trenches in 2 small marlous valleys, tributaries of the Peroulades river, C/ Pedo- and chronostratigraphical contextes of the Sidari 1 sequence with the main Neolithic levels (after Berger et al. 2014), D/ Pedo- and chronostratigraphical contextes of the Sidari 2 sequence with the main Holocene lithostratigraphic disconnexions.

Fig.11. A/ Mid-lower pedosedimentary sequence of Sidari 2 with early holocene paleosols (P1-P4), aggradation and gully phases (IIa-VIc). Yellow stars : AMS radiocarbon dates, B/ stairway Age depth model with three phases of high acceleration of sedimentation rate (phase II : 10.4-9.9 ka, phase IV : 9.5-8.9 ka, phase VI : 8.4-8.1 ka), C/Field photo of gravel and sand filling of the 9.3 event gullying, D/ Field photo of the 9.3 event gully filling with numerous rounded clay aggregates eroded in the upper catchment, E/ Field photo of the upper part of the 8.2 ka event filling with a regular alternation between silty and sandy beds,  F/CPDF of Sidari 1 and 2 sites (33 AMS dates). Paleosols are located





between main active peaks. Archaeological layers are represented as temporal coloured segments to
distinguish their cultural attribution.
Fig.12. A/ Comparison of regional hydroclimatic pattern for Anatolia and Northern Aegean areas with
micro-regional and main sites cumulative probability density : 1.Endorheic plains of central Anatolia
(Kuzucuoglu, this paper), 2. Gully erosion/fluvio-colluvial aggradation in Sidari 1/2 (Berger this
paper), 3. Soil formation in Sidari 1/2 (Berger this paper), 4.Lake Maliq Pollen-based temperature of
the coldest month (Albania, Bordon et al. 2009), 5. Oncoliths deposits in Dikili Tash swamp
(Macedonia, Lespez et al. this paper), 6. Detritism in Lake Dojran (Macedonia) (Zhang et al. 2014), 7.
Tenaghi-Philippon Tree-Pollen (%) (Macedonia, Pross et al. 2011), 8.Central Anatolia Late Neolithic
sites (Shortlived dates, n=123), 9. N.W. Turkey (shortlived dates, n=83), 10. Nea Nikomedia
(Macedonia)(12 shortlived dates) Pyke and Yiouni 1996, 11. Sidari (Corfu island) (12 charcoal, 3
shortlived dates) Berger et al. 2014 and in progress (RM: red monochrome ware, IP : Impressa ware),
12. Dikili Tash (11 charcoal dates) (Macedonia, Lespez et al. 2013). B/ Comparison of time dynamic
of Neolitisation from Central Anatolia to Corfu island. 1. Central Anatolia, All n=285, Shortlived
n=123 (after Flohr et al. 2015), 2. Western Anatolia  All n = 64, Shortlived n=31 (after Flohr et al.
2015), 3. NW Turkey, all n =136, shortlived n=83 (after Flohr et al. 2015),  4.Strong decline of site
occupation in Tell Sabi Abyad (North Syria) (from Weninger et al. 2014), 5. Paliambala (5 dates, after
Karamitrou-Mentessidi et al. 2013), 6. Nea Nikomedia, Thessalia (16 dates, Weninger et al.2006)(12
dates, "shortlived", Pyke and Yiouni 1996), 7. Mavropigi-Filotsairi, Macedonia (12 dates, after
Karamitrou-Mentessidi et al. 2013), 8. Sidari, Corfu island (15 dates) Berger et al. 2014 and in
progress (RM: red monochrome, IP : Impressa ware), 9. Dikili Tash, Macedonia (11 dates) (Lespez et
al. 2013), 10. Achilleion, Thessalia (44 dates) (B. Weninger et al. 2006).
fig. 13. Morpho- and pedosedimentary contextes of 4 Central to Eastern Mediterranean Early
Neolithic sites (Konispol cave, Sidari, Dikili Tash and Khirokitia) illustrating the 8.2 ka event effects
on the archaeological occupations. Geomorphological change applies on pure anthropogenic horizons
or paleosols, revealing an abrupt change of the local pedosedimentary functioning. 1. Gravels layer, 2.
sandy layer, 3. silty layer, 4. ashy layer, 5.oncolithic sands, 6.paleosols, 7. In-situ Neolithic layers, 8.
Slighty reworked Neolithic layer, 9.strongly reworked Neolithic layer, 10. Red silty clay colluvial
deposit (from Terra Rossa), 11. flints/ceramics, 12. Earth. Radiocarbone dates are in ka cal. BP.
fig. 14. Neolithic dynamic and Early Holocene RCC in Anatolia. Note: Sites are selected on the basis
of being the oldest ones excavated in their region (ie, sites founded after 8.0 ka cal BP are not shown).
Sources: Fontugne et al. (1999), Kuzucuoğlu et al. (1997, 1998, 1999), Düring (2002, 2011), Boyer et
al., 2006, Gürel & Lermi (2010), Özbaşaran (2011), Baird (2012), several articles in Özdoğan et al.
(2012a, 2012b), Kuzucuoğlu (2013, 2014), Stiner et al. (2014).

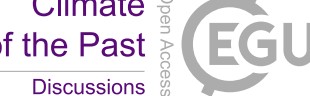

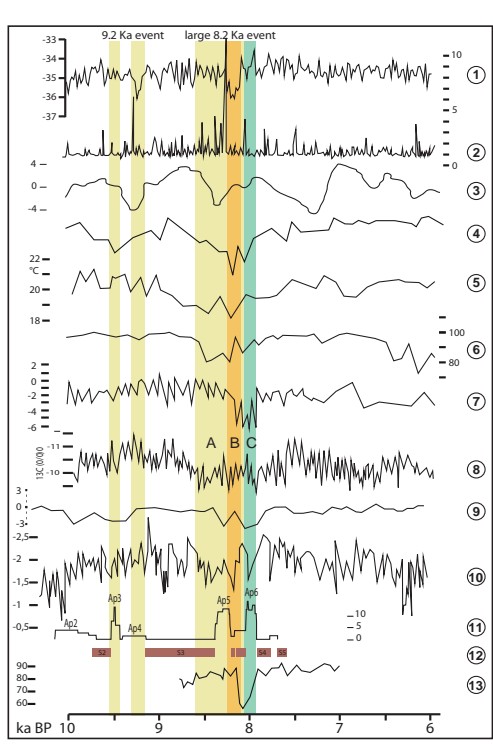

fig1



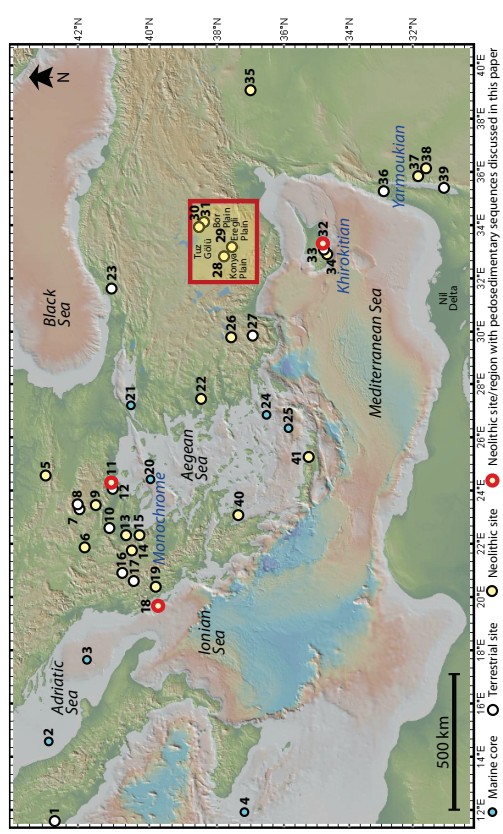

fig2

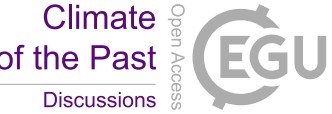







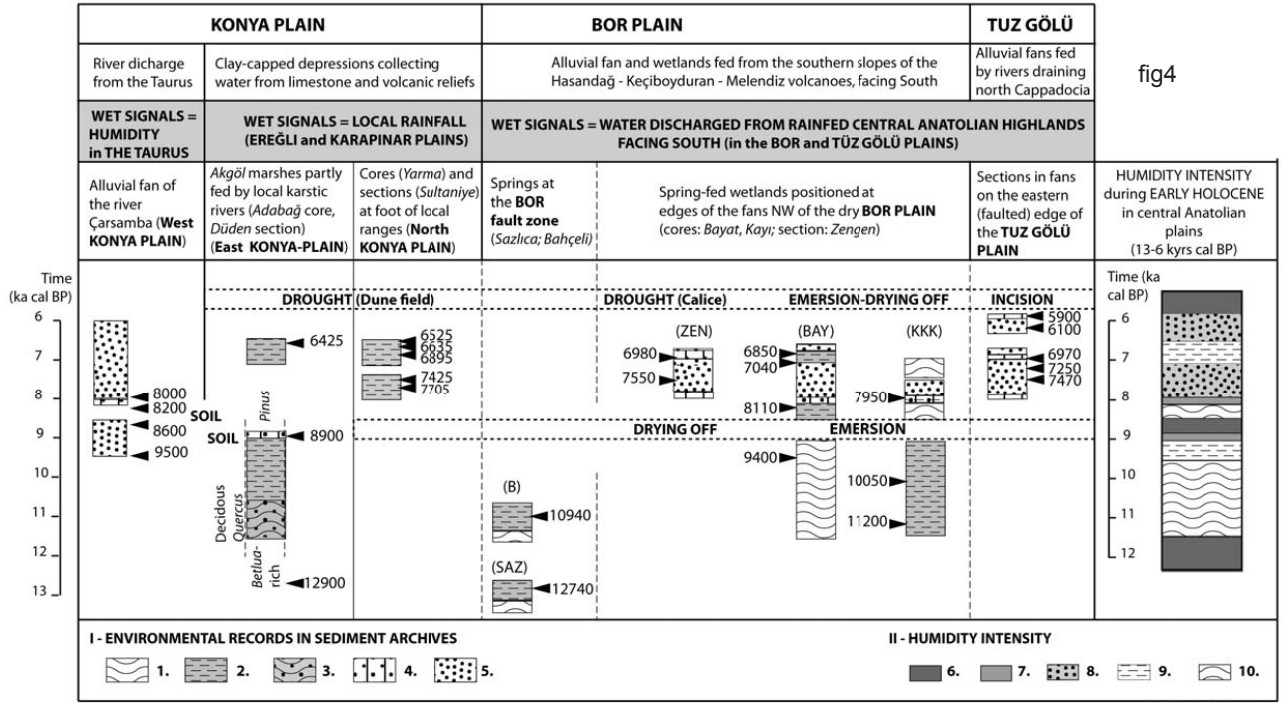

fig4

Sources are for (a) Konya and Ereğli (Bottema & Woldring, 1984; Kuzucuoğlu et al., 1997, 1998, 1999, in prep.; Fontugne et al., 1999; Boyer et al., 2006);
(b) Tuz Gölü (Naruse et al., 1997; Kashima, 2002)
Radiocarbon dates in Konya are from LSCE . AMS radiocarbon dates in Bor are from ARTEMIS/Saclay and POZNAN (PaleoMex/ArchéoMed)





fig5





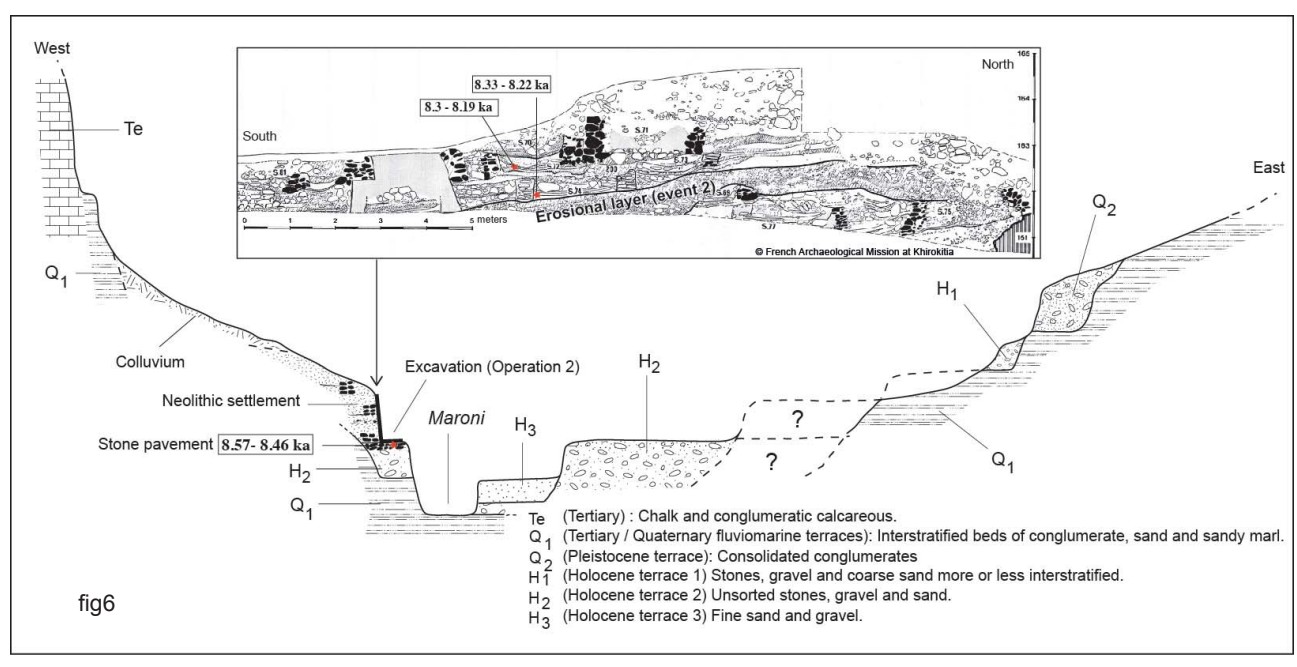





Dikili Tash

Dik 4 ●
*Glais et al. (2016)*

*Marsh of
Tenaghi-Philippon*

Greig and Turner (1974)

Pross *et al.* (2007 ; 2009 ; 2015)
Müller *et al.* (2011)
Peyron *et al.* (2011)
Tzedakis *et al.* (2006)

Wijmstra (1969)
Tzedakis *et al.* (2006)

Pross *et al.* (2015)

Greig and Turner (1974)

0                    2,5                    5 km    Image © 2015 DigitalGlobe



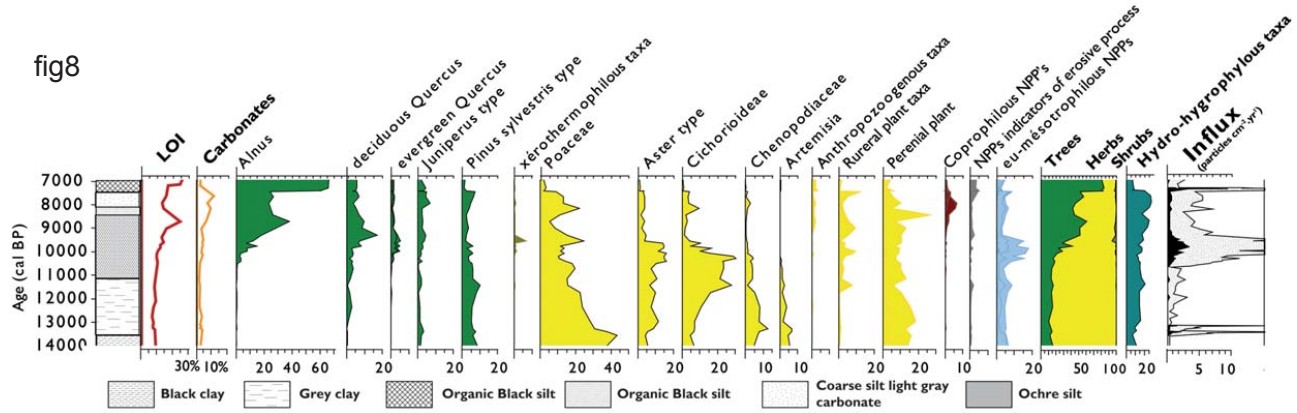

fig8





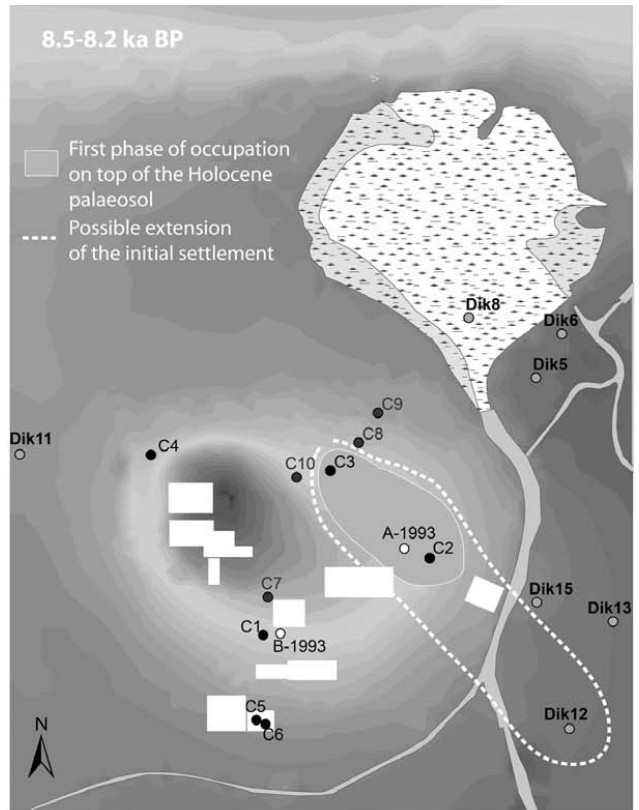
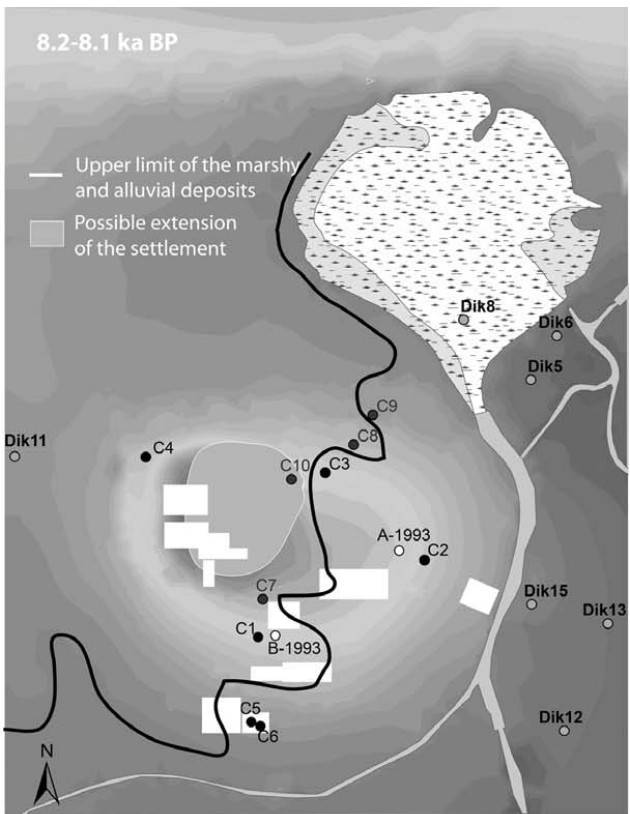
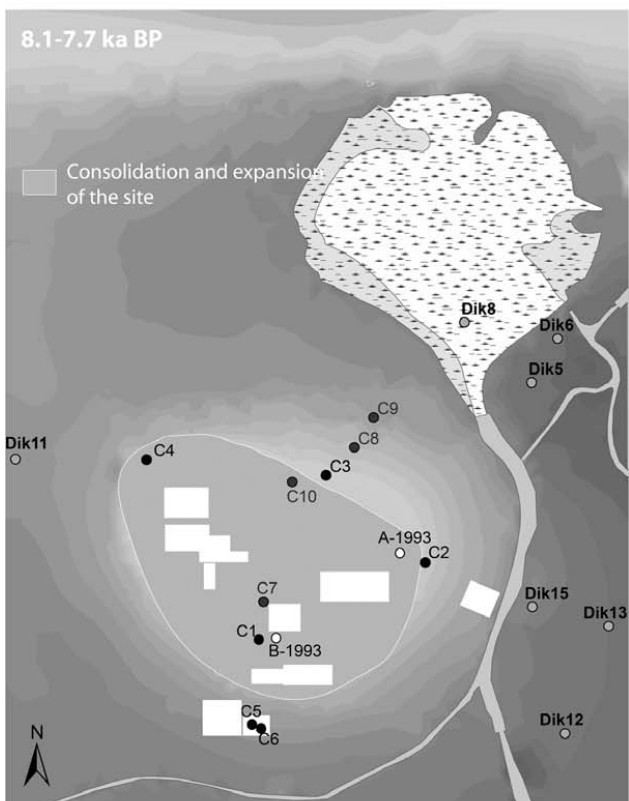
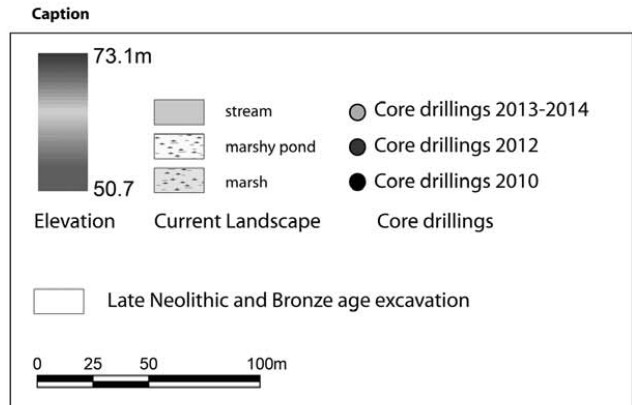

fig9





fig10

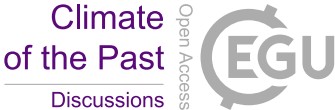



fig11

S : soil, Ap : Aggradation phase



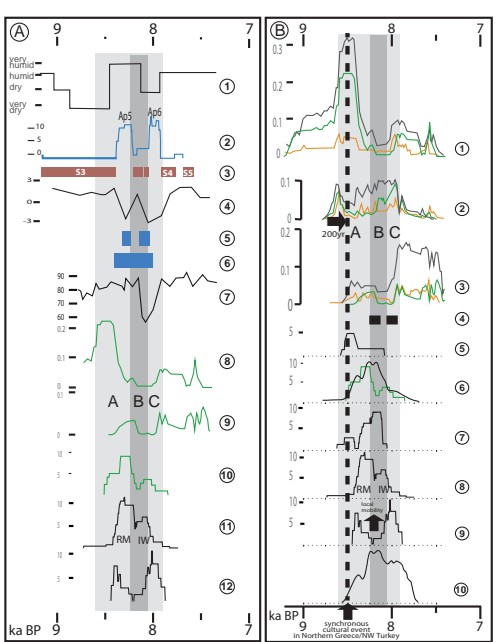

fig12



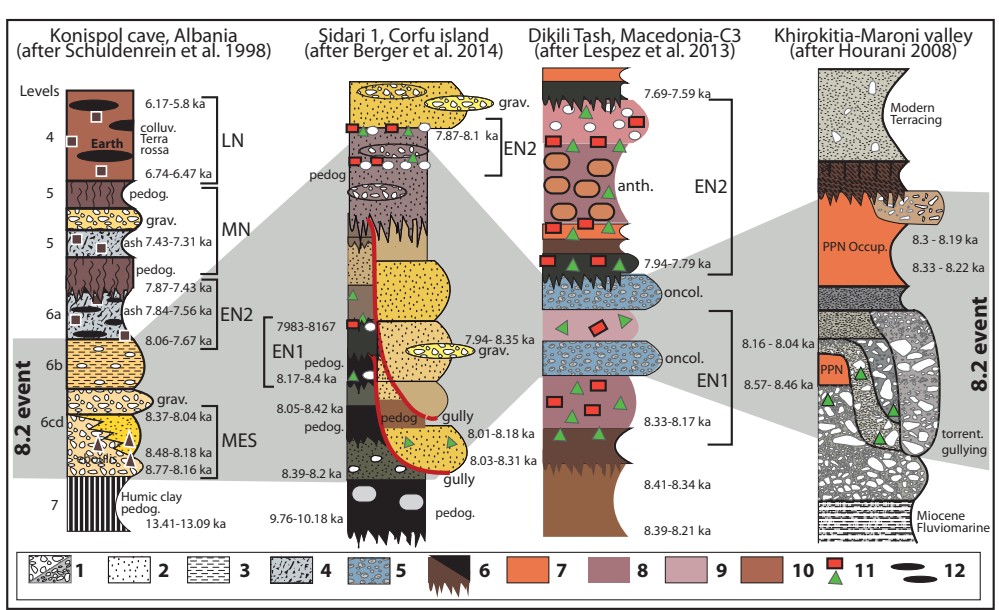

fig13





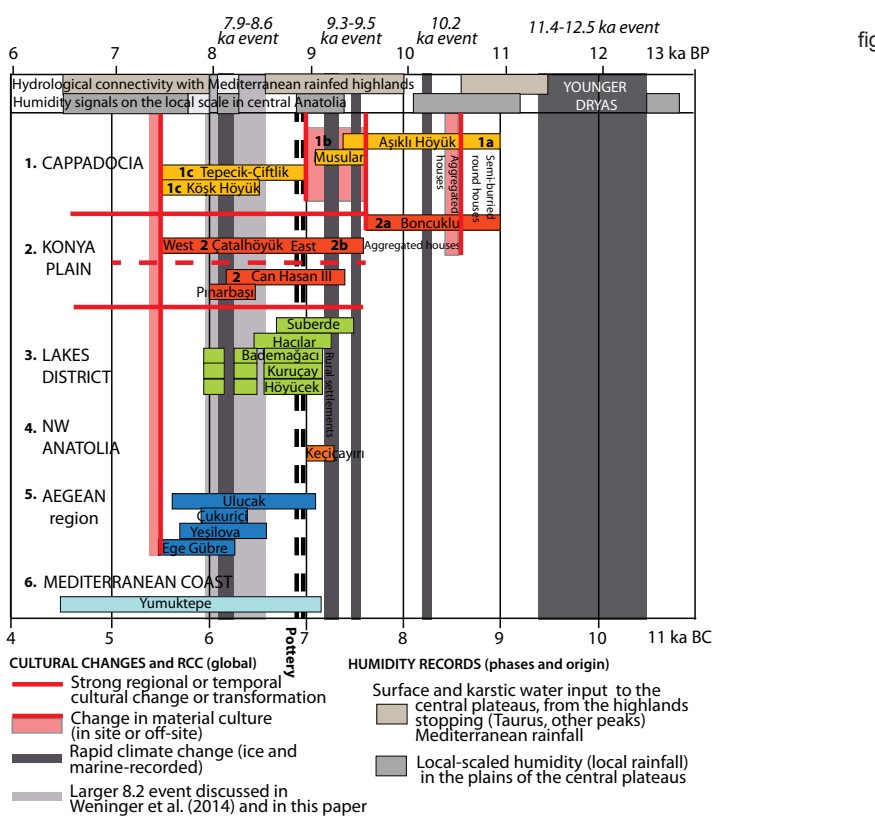