# Peer review of "Interactions between climate change and human activities during the Early to Mid Holocene in"

_Climate of the Past, 2016_

## Referee Comment (RC1) · Anonymous Referee #1 · 22 Mar 2016

General remarks:

In this paper, the authors study Early Holocene Rapid Climate Change events on a local scale by using new as well as earlier published geomorphological and other relevant terrestrial and lake data. Such a focus on local, sensitive archives that can be linked to nearby archaeological sites is extremely welcome and complements the existing regional and larger scale records well. Especially strong are the case studies that directly link the geomorphological record and archaeological sites, e.g. at Khirokitia. That local records are important is for example shown by apparently contradictive results from the Greece/Balkans records in this paper compared to regional studies; as the authors explain, the local water balance is not always reflecting the regional

climate trend (line 664-5). As such, the paper addresses relevant scientific questions in a novel way, reaching important conclusions. I therefore recommend that this paper is published in Climate of the Past.

I have some minor suggestions to propose though, as outlined below. Firstly, it would be extremely useful to have a description of the methods that were used. The material and methods section currently gives the cultural background, which is certainly useful, but in addition the more 'technical' methods should be described. This section should also clarify why certain methods are used in some of the regions, and others methods in others (e.g. pollen vs geomorphology). Was this what was available or a more conscious choice? The methods section could further include a description of the sites, which is currently partly present in the results section (e.g. line 308-311), but which I would move to the methods section. It could also explain why certain records were chosen but not others (e.g. Dik4 and 12, but not the other cores visible on Fig9). I would also move line 517-520 to the methods sections, where it can then also be explained into more detail.

Secondly, considering the importance of chronology as also stressed by the authors, it is important to present the chronology of the new records in detail and discuss the reliability of the chronology of the existing records and archaeological sites that are used in the paper (e.g. line 318). How were they dated, on what material, what were the dating uncertainties? Dates that have not been published before should be presented in a table and ideally also as age models (this could be done in the Supplementary Information). I am referring here for example to the Dik4 and 12 cores (line 445), NW Greece (line 515), and the site of Sidari (line 518; an age model is already presented in Fig11 and is very helpful). For the dates that were published elsewhere already, it would still be useful to have an overview presented here too.

Specific feedback:

Duration of the 8.2 ka event, line 105-111: This needs some clarification. It now seems

from the text that this may be the same event which is either short or long. However, it is also possible that there was both a more general cooling from 8.6 to 8.0/7.8 ka BP caused perhaps by an enhanced SH, and a more extreme, shorter (more rapid) cooling superimposed on this from 8.2-8.0 ka, caused by a meltwater spike. While this is something that is under discussion, it is necessary, in my opinion, to give this possibility in the paper too.

Discussion: Line 720-721: Some of these early dates are (yet) uncertain as they are in some cases based on charcoal (old wood effect possible, e.g. the earliest Ulucak dates). Line 723-725: how does the appearance of pottery increase the capacity to travel? Previously, other types of containers would have existed (e.g. made of skin, these could also have been decorated), which would in fact be more practical to travel with than heavy pottery.

Conclusion: The future directions are very useful, but it would also be good to have more of a summary/set of clear conclusions coming forth from the paper.

The discussion on synchronicity is generally very considered. In line 233-234 I wonder though: are these really synchronous? At 9.6/9.5 they seem too late for 10.2 and too early for 9.2?

Line 298: Baird 2012 (and other authors) identifies a distinct break at or after 8000 BP.

Line 308-309: The Tigris headwaters are much further east - I would leave this sentence out.

Technical corrections:

Line 14: remove "s" in "Changes"

Lines 26 and 649: Sapropele should be Sapropel

Line 30: I would replace "RCC is" by "Rapid Climate Changes (RCCs) are" to introduce the term clearly (in case the reader has not read the abstract first). It would also be

good if RCC were to be defined, as different interpretations of what "rapid" constitutes are in use in the literature.

Line 39-40: "the vicinity...at site." I think climate and environmental investigations are meant - could this be rephrased to make this clearer? Also, this section refers to "ongoing researches" (line 35) and as such references need to be specified.

Line 45: "chronology" should be "chronological"

Line 78: replace "rhythmed" by "interspersed" or similar. I would also specify what these pulsations entailed (i.e. it became colder/more arid).

Line 95-96: "The potential...been explored." Very true, but please insert references that have looked into this (e.g. Borrell 2007 in Neo-Lithics, Flohr et al. 2015).

Line 136: what is meant by secular? Best to replace this word, for example by centennial (if this is meant?). See also line 601.

Line 158: "in" should be "to"

Line 165: Replace "shortlived" by, for example, "radiocarbon" (not only shortlived dates were used). The point is a very good one though, systematic intra-site evaluations are essential.

Line 185 and 187: "as far as...is concerned" is repeated

Line 222: this is the Early PPNB, not Middle

Line 237: "this" refers to Musular, not Catalhoyuk - please rephrase

Line 257: "records that" - is a word missing here?

Line 259: remove "n" from "Anatolian"

Line 298: "many archaeologists" - this needs references.

Line 302: To be consistent, it would be good to use either 9.3 or 9.2 throughout the

manuscript.

Line 435: "research" - should be something in plural, e.g. "studies"

Line 501: should be "of the Central"

Line 503: considering replacing "important"

Line 510: insert comma in between "floor present"

Line 513: is the figure number correct?

Line 605: remove "largest"

Line 606: remove "was" and "by"

Line 788: "Episode" - should this be "This episode is"?

Line 789: should be "of the 8.2"

Fig. 5: What is MI?

---

## Referee Comment (RC2) · B. Paolo (Referee) · 13 May 2016

The paper is a major contribution to the study of the cultural and climatic changes that took place in the Eastern and Central Mediterranean region during a key period of prehistory. Though based on the study of only four main areas, the paper is extremely important and should be published as it is. The notes that follow might contribute to the general discussion and migh be utilised in some cases according to the will of the authors.

Line 283: I think that a reference is necessary for Odmut Cave. See for instance Kozlowski et al 2004 published in Warsaw (1994)

2.2. At this point I would mention the importance of Mavropigi and the westernmost part of Western Macedonia as a whole. See also below.

Line 610: There is a paper with important data (especially at page 26) on the Arabian Sea monsoons and their effects. See Yoganandan V. et al 2013 In Climate Change and Island and Coastal Vulnerability, pages 21-30. This might be mentioned.

4.1 heading. I would put somewhere under this heading a few words about Dispiliò (Lake Kastorià) from which we have lots of alomst unknown known archaeological data. It is the only Greek pile dwelling excavated in a proper way. Although mostly Late Nelithic the pollen cores etc.. would cover a lso earlier periods.

Line 693, following: I think that the same instabity recurs also from the Trieste and Istrian Peninsula caves. See cave Edera for instance. This is a fact that makes the Early Holocene and especially the Early Atlantic sequences of some regions of the Balkans and the Alpine arc of problematic interpretation. Little work has ever been carried out on the topic. The same can also be observed in the long sequences of the Crimean mountains Shan-koba, Murzak-koba etc..) See the papers by Cordova on the pollens and forest covers in the CRimean mountainsand a rediscussion of the sequence in Biagi (Tuebingen 2016)

Line 748: I would mention here the Western Macedonian site Mavropigi and the others in the region. The Mavropigi radiocarbon dates have been published recently on Antiquity Projects and also in Eurasian Prehistory 2015. This is a key area for the interpretation of the origin of the south-west Balkan and also the Adriatic Neolithic as a whole (in my opinion). Most data are unpublished unfortunately.

Another important topic is the little knowledge we have of the Late Mesolithic in the Balkan Peninsula as a whole: its origin and its end. Almost no step forward since the seminal paper by JGD Clark, 1958, on the Blade and Trapeze asssemblages. We know almost nothing of the Early Atlantic period in the entire Balkan Peninsula (and Greece). Are the many radiocarbon dates from very restricted regions, like the Iron

Gates for instance, mostly unrelated to well defined cultural complexes so important in this respect??

---

## Author Comment (AC1) · 23 Jun 2016

General remarks:
In this paper, the authors study Early Holocene Rapid Climate Change events on a local scale by using new as well as earlier published geomorphological and other relevant terrestrial and lake data. Such a focus on local, sensitive archives that can be linked to nearby archaeological sites is extremely welcome and complements the existing regional and larger scale records well. Especially strong are the case studies that directly link the geomorphological record and archaeological sites, e.g. at Khirokitia. That local records are important is for example shown by apparently contradictive results from the Greece/Balkans records in this paper compared to regional studies; as the authors explain, the local water balance is not always reflecting the regionalclimate trend (line 664-5). As such, the paper addresses relevant scientific questions in a novel way, reaching important conclusions. I therefore recommend that this paper is published in Climate of the Past.

I have some minor suggestions to propose though, as outlined below. Firstly, it would be extremely useful to have a description of the methods that were used. The material and methods section currently gives the cultural background, which is certainly useful, but in addition the more 'technical' methods should be described. This section should also clarify why certain methods are used in some of the regions, and others methods in others (e.g. pollen vs geomorphology). Was this what was available or a more conscious choice?
A new paragraph was added to better present field approaches and analytic methods used. We explain in particular why the pollen studies have not been common to all of the sites studied (opportunity and taphonomy).

The methods section could further include a description of the sites, which is currently partly present in the results section (e.g. line 308-311), but which I would move to the methods section. It could also explain why certain records were chosen but not others (e.g. Dik4 and 12, but not the other cores visible on Fig9). I would also move line 517-520 to the methods sections, where it can then also be explained into more detail.
A more detailed description of the sites has been made. It is partly based on a reorganization of the text, as requested by the reviewer. We explain in the text why Dik4 and 12 were choozen, only because these were the only performed in PVC tubes.

Secondly, considering the importance of chronology as also stressed by the authors, it is important to present the chronology of the new records in detail and discuss the reliability of the chronology of the existing records and archaeological sites that are used in the paper (e.g. line 318). How were they dated, on what material, what were the dating uncertainties? Dates that have not been published before should be presented in a table and ideally also as age models (this could be done in the Supplementary Information). I am referring here for example to the Dik4 and 12 cores (line 445), NW Greece (line 515), and the site of Sidari (line 518; an age model is already presented in Fig11 and is very helpful). For the dates that were published elsewhere already, it would still be useful to have an overview presented here too.
A new paragraph is dedicated to the presentation of the absolute radiocarbon data. It refers to unpublished data and previously published data. A table was added in supplt material with data from Sidari, Dikili Tash and Khirokitia sites with all the informations requested. The more regional Anatolia data, based on many previous works, simply refer to the publications. An age model of the DIK4 core is now available in the fig.8.

Specific feedback:
Duration of the 8.2 ka event, line 105-111: This needs some clarification. It now seems from the text that this may be the same event which is either short or long. However, it is also possible that there was both a more general cooling from 8.6 to 8.0/7.8 ka

BP caused perhaps by an enhanced SH, and a more extreme, shorter (more rapid) cooling superimposed on this from 8.2-8.0 ka, caused by a meltwater spike. While this is something that is under discussion, it is necessary, in my opinion, to give this possibility in the paper too.

It seems that this is exactly what we are discussing in this paper, particularly by relying on two papers (Weninger et al. 2014 and Rohling & Pälike 2005). The recent extension of the duration of the 8.2 ka event to a longer 8.6-8.0/7.9 ka period, due to the arrival of new data in the field of Neolithic Anatolia and northern Greece, in previously poorly explored regions, may explain this new positioning which calls into question the initial deterministic theories. We draw up a table of all published data or new data in archaeological context. Weninger et al. propose 2 successive rapid changes. We observe a temporal tripartite division with a probable climate improvement between the two phases of pejoration. This hypothesis is currently being explored and validating.

Discussion: Line 720-721: Some of these early dates are (yet) uncertain as they are in some cases based on charcoal (old wood effect possible, e.g. the earliest Ulucak dates).

We add in the texte that early dates ca 9. ka cal BP are waiting multiplication for being representative.

Line 723-725: how does the appearance of pottery increase the capacity to travel? Previously, other types of containers would have existed (e.g. made of skin, these could also have been decorated), which would in fact be more practical to travel with than heavy pottery.

We delete this sentence devoted to a technical aspect, which does not appear fundamental in our demonstration.

Conclusion: The future directions are very useful, but it would also be good to have more of a summary/set of clear conclusions coming forth from the paper.

2 sentences are added to summary the main results.

The discussion on synchronicity is generally very considered. In line 233-234 I wonder though: are these really synchronous? At 9.6/9.5 they seem too late for 10.2 and too early for 9.2?

Yes, we agree of course but we do not try here to match this cultural change (emergence of the Khirokitia culture) with one or other of these RCC. Our sentence was perhaps clumsy. We specified that « If all the changes identified on the Cypriot PPNB sites can not all be discussed at the moment in relation to the 9.2 ka event, due to lack of a clear temporal synchronization ; the question of climate control can be asked in Cyprus on economic transition towards pastoralism published by Vignes et al. (2011). This assumption for example, was recently proposed for the near east by Flohr et al. (2015).”

Line 298: Baird 2012 (and other authors) identifies a distinct break at or after 8000 BP.

Sentence completed in the texte with references to Baird 2012, Marciniak et al., 2015 and Düring, 2011.

Line 308-309: The Tigris headwaters are much further east - I would leave this sentence out.

Geographic paragraph from "The wide and endorheic plains of central Anatolia" moved in the "material and method" part, and the sentence about Tigris is leaved out.

Technical corrections: all made and completed in the texte
Line 14: remove "s" in "Changes"
Lines 26 and 649: Sapropele should be Sapropel
Line 30: I would replace "RCC is" by "Rapid Climate Changes (RCCs) are" to introduce the term clearly (in case the reader has not read the abstract first). It would also be good if RCC were to be defined, as different interpretations of what "rapid" constitutes are in use in the literature.

« Rapid Climate Change », often also called « Abrupt Climate Change », concerns most often a period of 150 to 400 years, that start abruptly within one decade or two at the most. Of course,

50 years are difficult to constrain with regard to uncertainties of the dating method mostly used in palaeoenvironmental and archaeological studies. This is why the following results and discussions stress the importance chronological resolution which has to be the highest possible, even in continental and/or disrupted contexts.

Line 39-40: "the vicinity: : :at site." I think climate and environmental investigations are meant - could this be rephrased to make this clearer? Also, this section refers to "ongoing researches" (line 35) and as such references need to be specified.

Line 45: "chronology" should be "chronological"

Line 78: replace "rhythmed" by "interspersed" or similar. I would also specify what these pulsations entailed (i.e. it became colder/more arid).

Line 95-96: "The potential: : :been explored." Very true, but please insert references that have looked into this (e.g. Borrell 2007 in Neo-Lithics, Flohr et al. 2015).

Line 136: what is meant by secular? Best to replace this word, for example by centennial (if this is meant?). See also line 601.

Line 158: "in" should be "to"

Line 165: Replace "shortlived" by, for example, "radiocarbon" (not only shortlived dates were used). The point is a very good one though, systematic intra-site evaluations are essential.

Line 185 and 187: "as far as: : :is concerned" is repeated

Line 222: this is the Early PPNB, not Middle

Line 237: "this" refers to Musular, not Catalhoyuk - please rephrase

Line 257: "records that" - is a word missing here?

Line 259: remove "n" from "Anatolian"

Line 298: "many archaeologists" - this needs references.

Line 302: To be consistent, it would be good to use either 9.3 or 9.2 throughout the C4. Homogenised in 9.2 in all the paper.

Line 435: "research" - should be something in plural, e.g. "studies"

Line 501: should be "of the Central"

Line 503: considering replacing "important"

Line 510: insert comma in between "floor present"

Line 513: is the figure number correct? No, of course. The good call is fig10cd

Line 605: remove "largest"

Line 606: remove "was" and "by"

Line 788: "Episode" - should this be "This episode is"?

Line 789: should be "of the 8.2"

Fig. 5: What is MI? I'm not find MI in the legende of the fig.5 ?

---

## Author Comment (AC2) · 23 Jun 2016

**Answer to reviewer 2 (P.Biagi)**

The paper is a major contribution to the study of the cultural and climatic changes that took place in the Eastern and Central Mediterranean region during a key period of prehistory. Though based on the study of only four main areas, the paper is extremely important and should be published as it is. The notes that follow might contribute to the general discussion and migh be utilised in some cases according to the will of the authors.

Line 283: I think that a reference is necessary for Odmut Cave. See for instance Kozlowski et al 2004 published in Warsaw (1994)
Reference added.

2.2. At this point I would mention the importance of Mavropigi and the westernmost part of Western Macedonia as a whole. See also below.
We recalled the importance of this site recently excavated in context of rescue archaeological excavation and its very old chronology at a regional scale.

Line 610: There is a paper with important data (especially at page 26) on the Arabian Sea monsoons and their effects. See Yoganandan V. et al 2013 In Climate Change and Island and Coastal Vulnerability, pages 21-30. This might be mentioned.
We have a look to this important paper for the Indian Ocean region (Yoganandan, V., Krishnaiah, C., Selvaraj, K., Prasad, G. R., & Dutta, K. (2013). Monsoonal Fluctuations vs Marine Productivity during Past 10,000 Years—A Study Based on Sediment Core Retrieved from Southeastern Arabian Sea. In *Climate Change and Island and Coastal Vulnerability* (pp. 21-30). Springer Netherlands), but the chronological precision of this marine core is not enough sufficient for our purposes, with 2/3 measures per millenium.

4.1 heading. I would put somewhere under this heading a few words about Dispiliò (Lake Kastorià) from which we have lots of alomst unknown known archaeological data. It is the only Greek pile dwelling excavated in a proper way. Although mostly Late Nelithic the pollen cores etc.. would cover a lso earlier periods.
Multi-proxies analysis has established a complex occupational pattern at the lakeside site of Dispilio from the Middle Neolithic to Chalcolithic Period, and  only a very small part of the littoral phase of the settlement has been excavated, but unfortunately the current chronology of this very interesting and promising site in not adequate with the chronology of our paper. We mention this site as having strong potential to advance in future research on palaeohydrological changes in North Greece in the 4.2. paragraph.

Line 693, following: I think that the same instability recurs also from the Trieste and Istrian Peninsula caves. See cave Edera for instance. This is a fact that makes the Early Holocene and especially the Early Atlantic sequences of some regions of the Balkans and the Alpine arc of problematic interpretation. Little work has ever been carried out on the topic.
We add that like in Caves and Rockshelter sites on the Tristine karst and in Istria, postdepositionnal processes mainly from anthropogenic origin are sometimes again into question. They show a temporal gap between the latest Mesolithic and the earliest Neolithic occupations (Mlekuz 2005, Forenbaher and Miracle 2005), but for them a lack of radiocarbon evidence, erosional surfaces by anthropogenic modifications and sedimentary hiatuses could explain these gaps, inversion of radiocarbone dates and presence of castelnovian microliths in Neolithic deposits (Mlekuz et al. 2008).

The same can also be observed in the long sequences of the Crimean mountains Shan-koba, Murzak-koba etc..) See the papers by Cordova on the pollens and forest covers in the CRimean mountains and a rediscussion of the sequence in Biagi (Tuebingen 2016)
In the paper of Cordova, C. E., & Lehman, P. H. (2005). Holocene environmental change in southwestern Crimea (Ukraine) in pollen and soil records. *The Holocene*, *15*(2), 263-277, we observe

precise climatological changes around 9.5-9.0 ka BP that represent an arid period.We add this information and reference in the paper.  But no precise data document the 8.2 ka event in Crimea on the 2 pollen series discussed in this paper. The radiocarbon dates series of Mesolithic sites seem indicate a hiatus in Shan-koba occupation around 8.4-8.0 ka, which confirms other regional data more to the West (Biagi 2016).

Line 748: I would mention here the Western Macedonian site Mavropigi and the others in the region. The Mavropigi radiocarbon dates have been published recently on Antiquity Projects and also in Eurasian Prehistory 2015. This is a key area for the interpretation of the origin of the south-west Balkan and also the Adriatic Neolithic as a whole (in my opinion). Most data are unpublished unfortunately.

We use in the fig.8 the currently published radiocarbone dates of this very important site, extensively excavated in a multi-proxy intra-site study, but as sublined by the reviewer, most data are still unpublished, and a part of radiocarbon dates are published in Greek.

Another important topic is the little knowledge we have of the Late Mesolithic in the Balkan Peninsula as a whole: its origin and its end. Almost no step forward since the seminal paper by JGD Clark, 1958, on the Blade and Trapeze asssemblages. We know almost nothing of the Early Atlantic period in the entire Balkan Peninsula (and Greece). Are the many radiocarbon dates from very restricted regions, like the Iron Gates for instance, mostly unrelated to well defined cultural complexes so important in this respect??

We are totally agree with this reflexion about Aegean and Balkan Mesolithic that one of us recently discussed (Berger in press). We know almost nothing of the Late Mesolithic (Blade and Trapeze assemblages) in the Balkan Peninsula. Only very restricted regions like the Iron Gates are documented, but far from the Egean coast. The same observation has also been done by Özdogan (2007) in western Turkey. The hiatus also seem to be bridged in part by systematic surveys as in the mountains of Pindus between Macedonia and Epirus (Efstratiou et al. 2006) or by geoarchaeological explorations further in deep floodplains and the vast sedimentary basins of the Aegean and Balkan world (Berger in press).